# Remote Sensing Monitoring of Rice Grain Protein Content Based on a Multidimensional Euclidean Distance Method

**Jie Zhang** [1,2], **Xiaoyu Song** [1,*], **Xia Jing** [2], **Guijun Yang** [1], **Chenghai Yang** [3], **Haikuan Feng** [1], **Jiaojiao Wang** [1] **and Shikang Ming** [1]

1   Information Technology Research Center, Beijing Academy of Agriculture and Forestry Sciences, Beijing 100094, China
2   College of Geomatics, Xi'an University of Science and Technology, Xi'an 710054, China
3   USDA-Agricultural Research Service, Aerial Application Technology Research Unit, College Station, TX 77845, USA
*   Correspondence: songxy@nercita.org.cn

**Abstract:** Grain protein content (GPC) is an important indicator of nutritional quality of rice. In this study, nitrogen fertilization experiments were conducted to monitor GPC for high-quality Indica rice varieties Meixiangzhan 2 (V1) and Wufengyou 615 (V2) in 2019 and 2020. Three types of parameters, including photosynthetic sensitive vegetation indices (VIs), canopy leaf area index (LAI), and crop plant nitrogen accumulation (PNA), obtained from UAV hyperspectral images were used to estimate rice GPC. Two-dimensional and three-dimensional GPC indices were constructed by combining any two of the three types of parameters and all three, respectively, based on the Euclidean distance method. The $R^2$ and RMSE of the two-dimensional GPC index model for variety V1 at the tillering stage were 0.81 and 0.40% for modeling and 0.95 and 0.38% for validation, and 0.91 and 0.27% for modeling and 0.83 and 0.36% for validation for variety V2. The three-dimensional GPC index model for variety V1 had $R^2$ and RMSE of 0.86 and 0.34% for modeling and 0.78 and 0.45% for validation, and 0.97 and 0.17% for modeling and 0.96 and 0.17% for validation for variety V2 at the panicle initiation stage. At the heading stage, the $R^2$ and RMSE of the three-dimensional model for variety V1 were 0.92 and 0.26% for modeling and 0.91 and 0.37% for validation, and 0.96 and 0.20% for modeling and 0.99 and 0.15% for validation for variety V2. These results demonstrate that the GPC monitoring models incorporating multiple crop growth parameters based on Euclidean distance can improve GPC estimation accuracy and have the potential for field-scale GPC monitoring.

**Keywords:** UAV; hyperspectral remote sensing; grain protein content; Euclidean distance; rice

## 1. Introduction

Rice is one of the three major crops in the world, with a wide distribution and long cultivation history [1]. As a main grain food crop, the rice planting area in China accounts for 20% of the world's planting area. Its output has been maintained at more than 200 million tons, accounting for nearly 40% of the world's total rice output. In recent years, with improvement of the living and consumption standards, people's demand for high-quality and good-taste rice grows rapidly in China. Rice with good tasting quality is an important factor in determining its market price. So, more farmers focus to improve the rice quality in order to enhance their income from rice planting. The grain protein content (GPC) of rice is an important factor affecting its nutrition and taste [2–4].

Remote sensing can be used to rapidly estimate crop nitrogen content and GPC. Some studies predicted wheat GPC through remote sensing data based on the theory that crop canopy spectral information can reveal the crop growth and nutrition status and which, in turn, can be used to guide the fertilizer management and adjust the GPC and improve cereal quality [5,6].





Numerous previous studies have focused on establishing the relationship between crop nitrogen status and spectral information. Wang et al. [7] used correlation analysis and Gaussian process regression (GPR) methods to select nitrogen sensitive spectra from the leaf and canopy level at different growth stages in rice. Yao et al. [8] found that the sensitive spectral band of leaf nitrogen accumulation is mainly located in the visible and near-infrared (NIR) portions by constructing a normalized difference spectral index (NDSI) and ratio spectral index (RSI) using different band combinations. Some studies that estimate GPC via remote sensing revealed the relationship between crop GPC and spectral indices [9–12]. Zhang et al. [13] used principal component analysis to spectrally downscale rice hyperspectral data. They selected the top four principal components to construct a rice GPC monitoring model. The correlation of the validated data based on different regression methods was greater than 0.9, indicating that the monitoring of grain crude protein content can be performed using canopy spectral data. Bagchi et al. [14] found that the improved partial least-squares method is most suitable for monitoring the straight-chain starch and protein contents of brown rice. Liu et al. [15] analyzed the hyperspectral characteristics of crude protein, crude starch, and amylose in rice. They considered that the absorption band of 2020–2235 nm was significantly correlated to rice crude protein and crude starch, with determination coefficients ($R^2$) of 0.639 and 0.884, respectively. Most current monitoring of GPC is based on ground-based hyperspectral or laboratory testing, which inevitably involves destructive sampling and is labor intensive in field surveys [16].

Satellite or airborne remote sensing systems can also be used to estimate crop nitrogen status and GPC for large areas [10,17]. However, acquiring airborne image data is usually expensive. Meanwhile the spatial resolution of satellite images is generally low, and cloud-free images can seldom be obtained during the cropping period in southern areas of China. Unmanned aerial vehicle (UAV) data, which has high temporal and spatial resolution and provides a high degree of versatility and flexibility compared to ground- measured and satellite data, may have the potential to improve interpretation of crop parameters [18–20]. UAV-based remote sensing techniques have been shown to be feasible for the estimation of crop indicators, such as leaf area index (LAI) [21], nitrogen [22], vegetation cover [23], and biomass [24] at the field scale. Compared to satellite images, UAV images can capture more accurate information of crop structure and greenness and facilitate the interpretation of crop growth and nutritional status. UAV-based multispectral and hyperspectral remote sensing have become important tools for precision crop management at the field level.

During the rice growth process, the formation of rice grain protein is often influenced by multiple factors [25,26]. The photosynthesis, canopy structure, and nutrient uptake affect the growth process of crop directly or indirectly. A single indicator often does not completely reflect crop growth. Therefore, it is necessary to consider the combined effects of different factors. The construction of vegetation indices (VIs) enhances the sensitive properties of some spectra and reduces noise from the soil or atmosphere [27], making it a more mainstream means of crop monitoring.

Chlorophyll content is a key factor in the photosynthetic capacity of vegetation and directly determines the ability of vegetation to respire and exchange energy with the outside world [28]. Chlorophyll or photosynthetic sensitive VIs, which are widely used in identifying the efficiency of photosynthetic light use in living plants, are good indicators of vegetation productivity and physiological status [29]. Crop plant nitrogen accumulation (PNA) is a direct indicator for plant nitrogen status during the crop growth processes [30]. The leaf area index (LAI) is one of the most important parameters of vegetation canopy structure, which can represent the crop growth conditions effectively [31,32]. The overall goal of this study was to estimate rice GPC using VIs related to chlorophyll and photosynthetic capacity, canopy structure factor, and crop plant nitrogen. The specific objectives were to: (1) select the appropriate hyperspectral indices for LAI and PNA inversion through arbitrary band combination methods, (2) use the Euclidean distance method to construct two-dimensional and three-dimensional GPC monitoring indices with different combinations of influence factors, and (3) evaluate the optimal GPC inversion model (i.e., established by appropriate

indices and algorithms) and generate field-scale rice grain protein monitoring maps by using the data collected from an agricultural experiment.

## 2. Materials and Methods

### 2.1. Experimental Design

Two rice nitrogen fertilizer gradient trials were conducted at Zhongluotan Experimental Station (23°23′24″N–23°23′59″N,113°25′48″E–113°26′24″E), Baiyun District, Guangzhou City, Guangdong Province, China, in the autumns of 2019 and 2020. The layout of the plots in the experimental field is shown in Figure 1. In 2019, the rice variety was Meixiangzhan 2 (V1) and the planting date was 8 August. Rice was planted manually at a density of 20 cm × 20 cm in 60 plots with a plot dimension of 3.2 m × 3.2 m or an area of 10.24 m$^2$. Only 15 plots were sampled and tested. The planting specifications were 16 × 16 clusters, making a total of 256 clusters/plot (Figure 1b). The rice varieties in the 2020 experiment were Meixiangzhan 2 (V1) and Wufengyou 615 (V2) and the planting date was 8 August. Manual planting was conducted at a density of 20 cm × 20 cm in 30 plots. The area of each plot was 13.12 m$^2$ (3.2 m × 4.1 m) and all plots were sampled and tested. The planting specifications were 16 × 20 clusters per plot, with a total of 320 clusters/plot (Figure 1c).

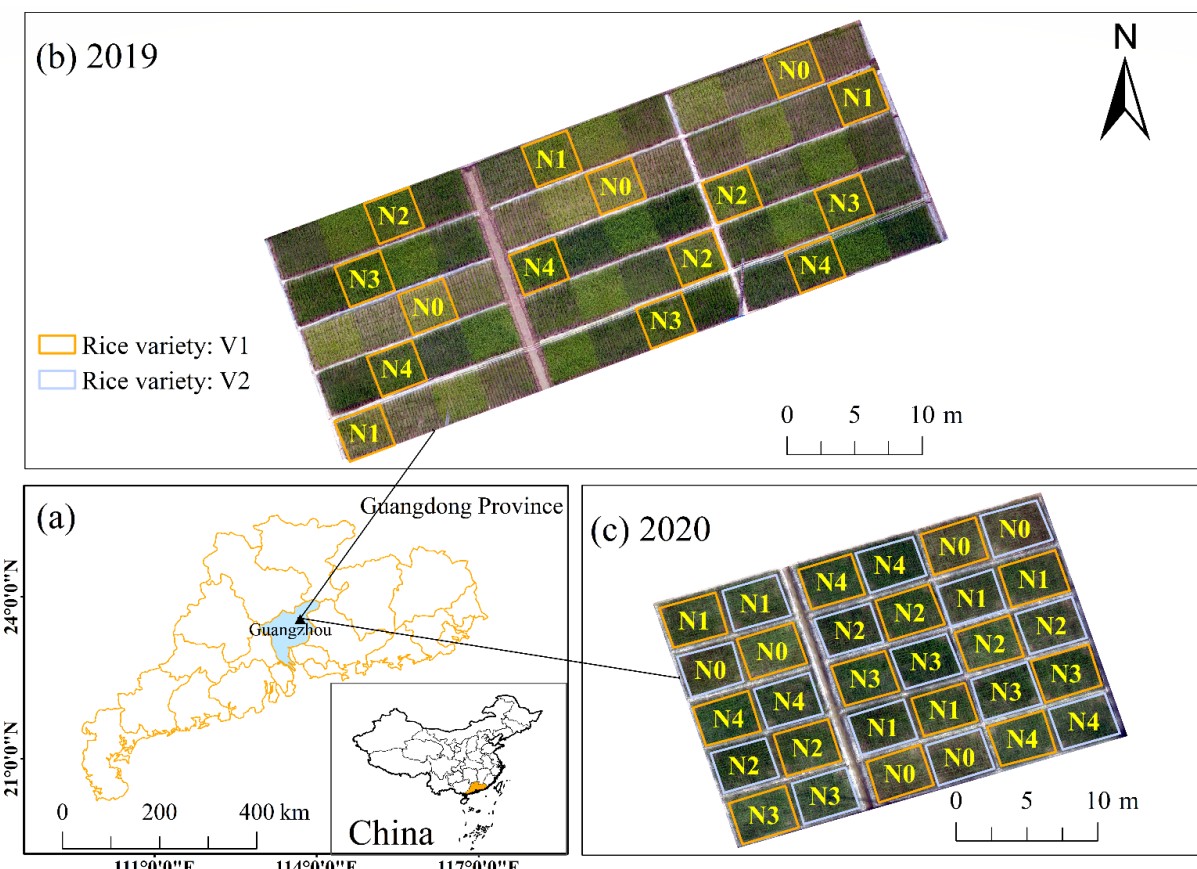

**Figure 1.** Study area and plot design for 2019 and 2020 experiments. (**a**) Location of test area, (**b**) Plot test map in 2019, (**c**) Plot test map in 2020.

Both trials used the same rice planting and fertilization management. Each experiment was designed with five nitrogen fertilization levels (N0, N1, N2, N3, and N4, corresponding to application rates of 0, 60, 120, 180, and 240 kg N/ha) with three replicates of each treatment. The ratio of basal fertilizer, tillering fertilizer, and spike fertilizer was 5:2:3. The amounts of phosphorus and potassium fertilizer used were 54 kg/ha and 144 kg/ha, respectively. The plots were separated by plastic film. A urea base fertilizer (46% N content) was applied at the seedling transplanting stage in the test plots at rates of 0, 65.2, 130.4, 195.7,

and 260.9 kg/ha for the five respective N levels; meanwhile, 120 kg/ha of potassium oxide (60% K content) was applied in all plots. Urea tiller fertilizer was applied at the tillering stage (mid-late August) at 0, 26.1, 52.2, 78.3, and 104.3 kg/ha for the five respective N levels. Urea spike fertilizer was applied at the rice panicle initiation stage (mid-September) at rates of 0, 39.1, 78.3, 117.4, and 156.5 kg/ha in the five respective N levels, while 120 kg/ha of potassium oxide (60% K content) was applied in all plots.

### 2.2. Data Acquisition

Field sampling and UAV data acquisition were conducted at the tillering stage (23 August, corresponding to Bundesanstalt, Bundessortenamt, Chemische Industrie (BBCH) phenological stage 23), panicle initiation stage (10 September, corresponding to BBCH 32), and heading stage (9 October, corresponding to BBCH-57) in 2020. UAV data and plant samples were only acquired at the panicle initiation stage (13 September, corresponding to BBCH 32) in 2019 [33,34].

### 2.2.1. UAV Data Collection

A six-rotor UAV (DJI S1000) equipped with an imaging spectrometer (Cubert UHD185 Firefly-type) was used to acquire hyperspectral images during the rice tillering, panicle initiation, and heading stages. Among them, the spectrometer had dimensions of 195 mm × 67 mm × 60 mm and was a full-frame, non-scanning, real-time imager that did not require IMU and post-data correction. The parameters of the sensor are shown in Table 1.

**Table 1.** Major specifications of Cubert UHD185 firefly imaging spectrometer.

| Parameters | Attributes |
| --- | --- |
| Place of origin | Germany |
| Weight | 0.47 kg |
| Spectral range | 450–950 nm |
| Spectral resolution | 8 nm@532 nm |
| Model | UHD185 |
| Spectral interval | 4 nm |
| Pixel | 1 million |
| Spatial resolution | 1.3 cm × 1.3 cm |

A whiteboard and a blackboard were used to radiometrically calibrate the UAV sensor before each flight. Flights were conducted at an altitude of 30 m and hyperspectral image cubes were captured at 5 frames per second, with an overlap rate of >50% per frame. The acquired hyperspectral data were converted from image DN values to reflectance values, and the reflectivity was calculated using Equations (1) and (2):

$$L^* = gain \cdot DN + bias \tag{1}$$

$$\rho^* = \frac{L^* \cdot d^2 \cdot \pi}{E_0 \cdot \cos \theta} \tag{2}$$

where DN is the grayscale value of the original image element, *gian* and *bias* are the corresponding gain and bias values of the sensor, $\rho^*$ is the apparent reflectance of the feature, $d$ is the solar–terrestrial astronomical unit, which is generally taken as 1, $E_0$ is the solar irradiance, and $\theta$ is the solar zenith angle.

Dense geographic point clouds, textured polygon models, and digital elevation models were generated from the overlapping images using Agisoft PhotoScan software (Agisoft LLC, St. Petersburg, Russia) to obtain orthomosaics of each rice growth stage. Finally, the vector of each plot was drawn in ENVI software, and an 80 cm × 80 cm area of interest was constructed in the center of each plot. The ENVI IDL program was used to extract reflectance data for each plot.

2.2.2. Plant Data Collection

1.    Rice LAI determination

LAI is the index of the total area of plant leaves per unit area to the total land area [35]. Samples of rice plants were collected from six clusters in each plot. A leaf area of about 0.2 m$^2$ (S) was measured and the leaves were dried and weighed ($w_1$). The rest of the leaves were also dried and weighed together ($w_2$). Then, the dry mass of leaves in the plot sample was $w_1 + w_2(W)$. *LAI* was calculated using the following equation:

$$SLA = LA/w_1 \tag{3}$$

$$LAI = 1/6 \times W \times SLA \times D \tag{4}$$

where *LA* is the measured leaf area (0.2 m$^2$), $w_1$ is the dry weight of leaves used to determine leaf area, $w_2$ is the dry weight of the remaining leaves, and *D* is the planting density, which was obtained by dividing the number of rice planting clusters in each small area by the plot area.

2.    Rice PNA determination

Six rice plant samples were randomly selected from each plot. Then the rice roots were removed by scissors and the numbers of stems and tillers counted manually. Rice leaves, stems, and spikes were separated and placed in an oven at 105 °C to dry green for 30 min firstly, then weighted after drying at 80 °C for 24 h. Then, the nitrogen contents for leaves, stems, and spikes were measured separately using a Kjeldahl nitrogen tester according to the equation:

$$NC = (V \times 0.05 \times 14 \times 1000)/(1000 \times M) \tag{5}$$

where *NC* is the nitrogen content (%), *V* is the volume of hydrochloric acid (mL), and *M* is the sample mass (g).

The biomass and nitrogen accumulation (*NA*) per unit area of leaves and plants were calculated based on the planting density and the dry weight of the rice samples according to the following equation:

$$PNA = (LAGB \times LNC + SAGB \times SNC + EAGB \times ENC)/100 \tag{6}$$

where *PNA* is the plant nitrogen accumulation (kg/m$^2$); *LAGB*, *SAGB*, and *EAGB* are the biomass (g/m$^2$) of leaves, stems, and spikes in the test samples, respectively; and *LNC*, *SNC*, and *ENC* are the N concentrations (%) of leaves, stems, and spikes, respectively. At the tillering and panicle initiation stages, only the relevant covariates of leaves and stems were calculated because the spikes were not yet developed.

3.    Rice yield and GPC determination

The yield was measured by harvesting 125 clusters of rice plants (5 m$^2$) for each plot, and then 100 g of rice grain was dried at 105 °C for 48 h to determine the moisture content of the grain. The rice yield for each plot was adjusted according to a moisture content of 14%. The remaining rice plants were threshed, and the seeds were dried for 3 months, hulled, and milled into fine rice, and then finely ground into flour. The grain nitrogen content was determined using the semi-micro Kjeldahl method, where GPC (%) = grain nitrogen content × 5.95 [36].

*2.3. Data Analysis Methods*

2.3.1. VI Calculation

Three VIs widely used in previous studies were constructed using the UAV reflectance data (Table 2) [16]. The MERIS terrestrial chlorophyll index (MTCI) and the red edge chlorophyll index (CI$_{red\ edge}$) are related to chlorophyll, and the photochemical reflectance index (PRI) is related to physiological status [37,38]. There are many factors affecting GPC synthesis in rice. In this study, for the sake of model universality and simplicity,

we selected several representative and highly recognized influencing factors in canopy structure, light energy utilization, and vegetation respiration as the construction parameters of the Euclidean distance method.

**Table 2.** Vegetation indices and formulas used in the study.

| Spectral Feature Type | Index Name (Abbreviation) | Index Formulation | Reference |
|---|---|---|---|
| Chl VI | MERIS terrestrial chlorophyll index (MTCI) | (R754-R709)/(R709-R681) | [39] |
| | Red edge chlorophyll index (CI$_{\text{red edge}}$) | (R800/R720)-1 | [40] |
| physiological VI | Photochemical reflectance index (PRI) | (R531-R570)/(R531+R570) | [41] |

### 2.3.2. Arbitrary Band Combination Spectral Index Calculation

According to the construction forms of common spectral indices, a sampling method was used to construct spectral indices of any two-band and three-band combinations. This allowed rapid feature extraction and analysis from a large sample of observation data and reduced the cost of feature selection [8]. Based on MATLAB programing, binary and ternary matrices were calculated for bands in the 454–950 nm interval, and each spectral band was added to the operation during the process. All possible VIs based on different band combinations were correlated with the LAI and PNA. The best VIs and spectral bands were obtained based on the two-dimensional and three-dimensional correlation coefficient matrices. The main reference forms included the ratio index (RVI), differential vegetation index (DVI), normalized difference vegetation index (NDVI), improved ratio vegetation index (IRVI), plant senescence reflectance index (PSRI), structure insensitive pigment index (SIPI), and arbitrary two- and three-band VIs according to the following equations:

$$RVI(\lambda_1, \lambda_2) = R_{\lambda 1}/R_{\lambda 2} \tag{7}$$

$$DVI(\lambda_1, \lambda_2) = R_{\lambda 1} - R_{\lambda 2} \tag{8}$$

$$NDVI(\lambda_1, \lambda_2) = (R_{\lambda 1} - R_{\lambda 2})/(R_{\lambda 1} + R_{\lambda 2}) \tag{9}$$

$$IRVI(\lambda_1, \lambda_2, \lambda_3) = R_{\lambda 1}/(R_{\lambda 2} + R_{\lambda 3}) \tag{10}$$

$$PSRI(\lambda_1, \lambda_2, \lambda_3) = (R_{\lambda 1} - R_{\lambda 2})/R_{\lambda 3} \tag{11}$$

$$SIPI(\lambda_1, \lambda_2, \lambda_3) = (R_{\lambda 1} - R_{\lambda 2})/(R_{\lambda 1} - R_{\lambda 3}) \tag{12}$$

where $\lambda_1$, $\lambda_2$ and $\lambda_3$ are wavelengths (nm); $R_{\lambda 1}$, $R_{\lambda 2}$ and $R_{\lambda 3}$ are reflectance corresponding to the wavelengths, and $R_{\lambda 1} \neq R_{\lambda 2} \neq R_{\lambda 3}$.

### 2.4. Construction of a GPC Monitoring Index Based on Euclidean Distances

Multiple factors that reflect crop growth conditions were considered in this study and expressed by the concept of Euclidean distance [42]. The Euclidean distance considers the real distance between two points in multidimensional space (Figure 2) and can be applied to various aspects, such as dryness monitoring [43] and remote sensing image processing [44]. The change in distance between zero point and other points in multidimensional space is influenced by multiple co-ordinate dimensions. This change in distance is regarded as the superposition of positive and negative factors in rice grain protein synthesis. Rice canopy spectral VIs expressing crop physiology and chlorophyll status, LAI representing canopy structure and phenotypic information, and PNA indicating the crop nitrogen nutrition condition were selected to construct the GPC monitoring index in this study. Two or three factors were combined to the GPC monitoring index by the Euclidean distance method and their accuracy for GPC estimation was investigated.

This study considered two- and three-dimensional monitoring indices constructed using two or three variables. The absolute distance between points in multidimensional space was measured by the Euclidean distance method to express crop growth under

multiple influences. As shown in Figure 2, each axis in multidimensional space was used to represent one variable affecting rice GPC, which was calculated as per Equation (13):

$$d(x,y) = \sqrt{\sum_{i=1}^{n}(x_i - y_i)^2} \tag{13}$$

where $d(x,y)$ is the Euclidean distance between $x(x_1, x_2, x_3, \cdots, x_n)$ and $y(y_1, y_2, y_3, \cdots, y_n)$, and n denotes the dimension of the space.

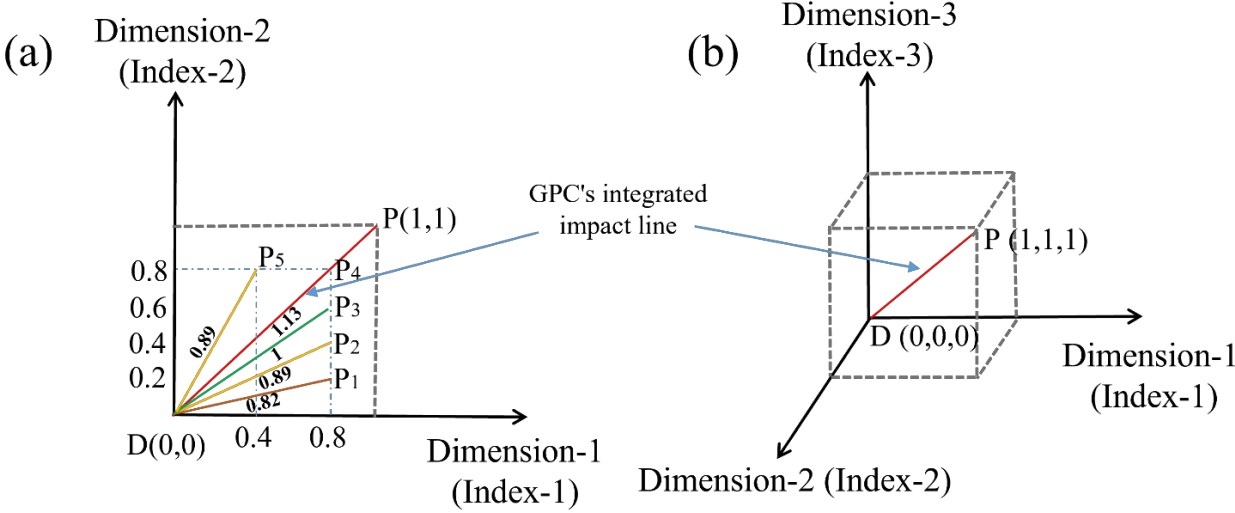

**Figure 2.** Schematic diagram of Euclidean distances in (**a**) a two-dimensional plane and (**b**) a three-dimensional space. The red line represents the distance between points D and P.

The minimum point D (0,0) in the space was chosen as the lowest point of GPC, and the distances from each point in the multidimensional space to this point were calculated. The distances from point D (0,0) to point $P_1$ (0.8,0.2) and to $P_5$ (0.4,0.8) in Figure 2a explain this variation. A greater distance indicates that the plants in this state could bring more positive effects on the formation of GPC. Conversely, a smaller distance means less favorable plant state for grain protein synthesis. This is because a single point does not necessarily control the distance from the point to the origin completely, but the distance is determined by the individual points together, which reflects the multi-factor joint action of rice GPC production. Considering that the Euclidean distance is affected by the dimension between parameters, the variables were normalized to the range between 0 and 1 using the following equation:

$$X_{\text{norm}} = \frac{X - X_{\text{min}}}{X_{\text{max}} - X_{\text{min}}} \tag{14}$$

where $X$ is the original data, $X_{\text{max}}$ and $X_{\text{min}}$ are the maximum and minimum values in the original dataset, respectively, and $X_{\text{norm}}$ is the normalized value.

### 2.5. Modeling and Accuracy Evaluation Methods

A multiple stepwise regression method was used to select the sensitive spectral indices obtained from the screening conducted by the arbitrary band combination approach. These selected indices were then used to construct LAI and PNA estimating models. The GPC monitoring index derived from Euclidean distances was used with linear, logarithmic, exponential, quadratic, and power functions to construct one-dimensional linear and nonlinear models of GPC. During modeling, two groups of nitrogen fertilizer gradient plots were randomly selected for modeling in 2019 and 2020, and the rest were used to test the accuracy of the models.

All the rice parameter models were evaluated for accuracy in terms of coefficient of determination ($R^2$) and root mean square error (RMSE), which are calculated as:

$$R^2 = 1 - {SSE}/{SST} \tag{15}$$

$$RMSE = \sqrt{\frac{1}{n}\sum_{i=1}^{n}(f_i - s_i)^2} \tag{16}$$

where *SSE* is the sum of the squares of residuals, *SST* is the sum of the squares of deviations, *n* is the number of samples, and $f_i$ and $s_i$ are the measured and predicted values, respectively.

## 3. Results

### 3.1. Analysis of Canopy Reflectance for Different Rice Varieties and Nitrogen Levels

Figure 3 shows the hyperspectral reflectance of the rice canopies of varieties V1 and V2 under different N treatments at the tillering stage (a, b), panicle initiation stage (c, d), and heading stage (e, f) in 2020.

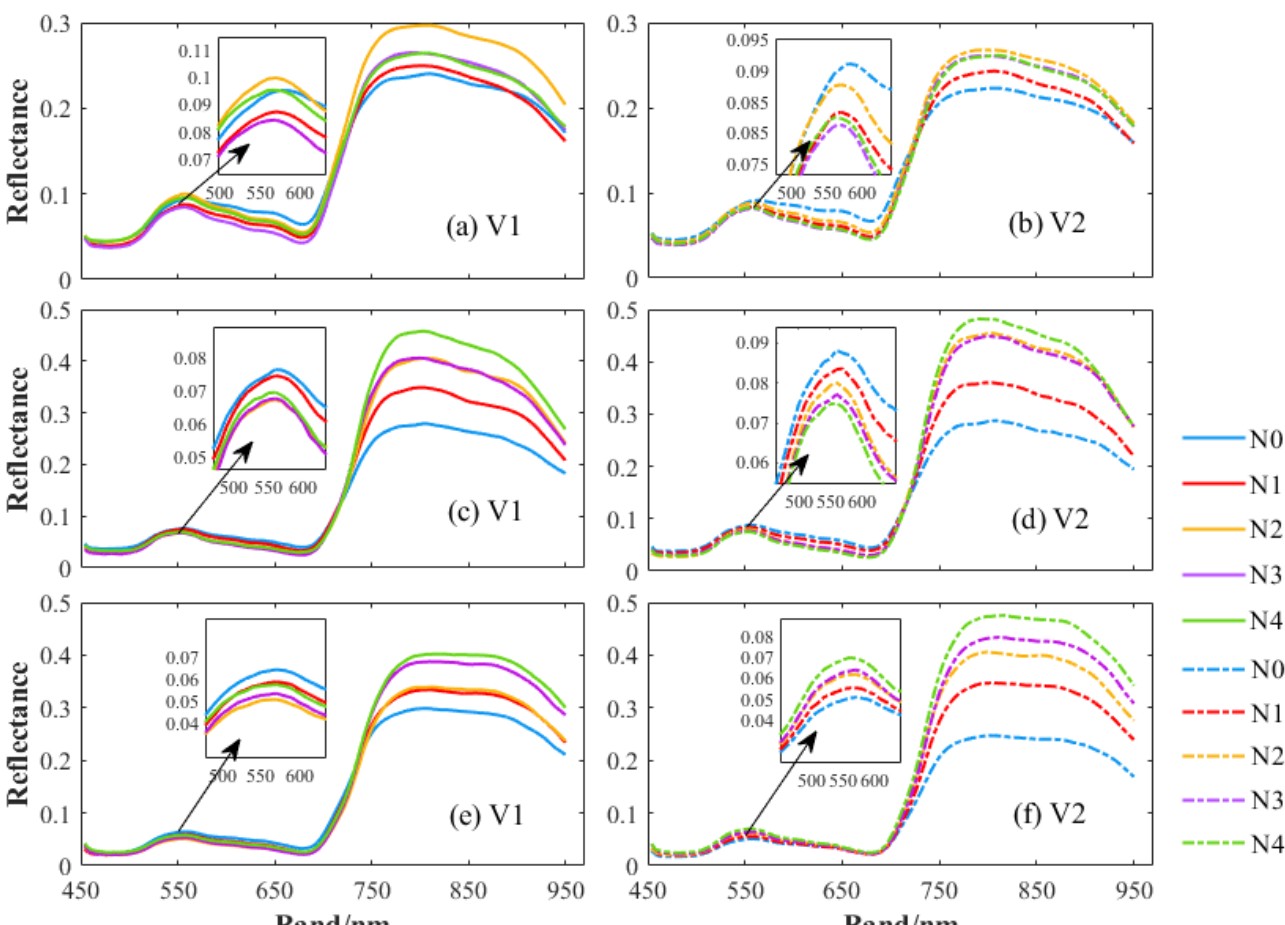

**Figure 3.** Canopy reflectance spectra of different rice varieties (V1 = Meixiangzhan 2 and V2 = Wufengyou 615) with different nitrogen fertilization levels. Reflectance of (**a**) V1 and (**b**) V2 at the tillering stage, (**c**) V1 and (**d**) V2 at the panicle initiation stage, and (**e**) V1 and (**f**) V2 at the heading stage.

It can be seen from Figure 3 that the N treatments affected the canopy reflectance in different rice growth stages. The overall trends in canopy reflectance in each stage were similar. The reflectance increased rapidly after 690 nm, resulting in a steep NIR shoulder. There was an obvious difference by nitrogen level, with the difference more obvious in

the NIR region than that around the green peak at 550 nm. Except for the variety V2 at the heading stage, all data showed that lower nitrogen levels were associated with higher reflectance around the green peak. This trend became more obvious as the growth stage progressed. The N0 treatment always had the lowest reflectance in the NIR region after 730 nm, while the N2 treatment had the highest reflectance at the tillering stage. During this stage, the rice plant is immature, and the rhizomes and leaves are in the developmental stage, which may inhibit the development of rice in cases of excess nitrogen. Meanwhile, the fertility period affects the growth and coverage of rice, and the spectra inevitably contain more background information when acquired during the tillering stage, which will affect the trend analysis to some extent. In contrast, during the panicle initiation and heading stages, when rice develops rapidly and tends to mature, the canopy cover increased and the reflectance of both varieties under treatment N4 was highest. There was generally an increase in reflectance with the amount of nitrogen applied, but the differences were small between some nitrogen treatment classes. For example, there were small differences between treatments N3 and N4 at tillering and between N2 and N3 at panicle initiation for V1 and V2, and between N1 and N2 at heading for V1. The other data showed that nitrogen application was the main influence on reflectance. Unlike leaf spectra, canopy spectra were somewhat more influenced by background factors, which would reduce the differences between adjacent nitrogen treatments. Comparing varieties V1 and V2, there were still some differences in reflectance at the same nitrogen level in the same growth stage. The reflectance of V1 at the tillering stage was higher than that of V2, but the difference was small. In the panicle initiation stage, V2 had higher reflectance. With the progress of the growth stage, this difference was more obvious in the NIR region. The reflectance difference under the same N level was close to 0.1. The reflectance in this region mainly comes from the internal vegetation structure. Different varieties will have different internal structures. Hence, there may be differences in sensitive areas when using reflectance data to retrieve agronomic parameters of different varieties of crops.

*3.2. Agronomic Parameters and GPC Analysis of Different Rice Varieties*

Figure 4 shows the distribution of LAI and PNA for the two rice varieties in the three growth stages in 2020. The LAI was higher in V2 at the tillering stage, and higher in V1 at the panicle initiation and heading stages, although the difference was small. The PNA was higher in V2 at the tillering and heading stages, with mean values about 0.3 and 1.8 higher than those in V1, respectively. The LAI and PNA of both varieties were closer at the panicle initiation stage. At this stage, rice enters a period of concurrent vegetative growth and reproduction and absorbs a large amount of nutrients from the soil. The biomass and N are stored in the stems and leaves, and the plant development approaches maturity so that differences between the varieties become smaller. In the heading stage, in addition to the stems and leaves, the spikes also store some dry matter, so the PNA content increases more than in the panicle initiation stage. The leaf area also increases as the reproductive stage advances, but not as significantly as in the tillering to panicle initiation stages. For example, the mean LAI of V1 increased from 0.62 in the tillering stage to 3.07 in the panicle initiation stage, but only from 3.07 to 4.22 between the panicle initiation and heading stages. This is mainly because the plants approached vegetative maturity from the tillering to panicle initiation stages and the absorbed nutrients are used for stem and leaf growth. After the panicle initiation stage, the plants tend to mature, the leaves gradually stop growing, and more nutrients are allocated to reproductive organs. Figure 4g shows that V1 ended up with a slightly higher GPC than V2, with a difference of about 0.1%, while there was a large difference of 3.26% between the different nitrogen treatments.

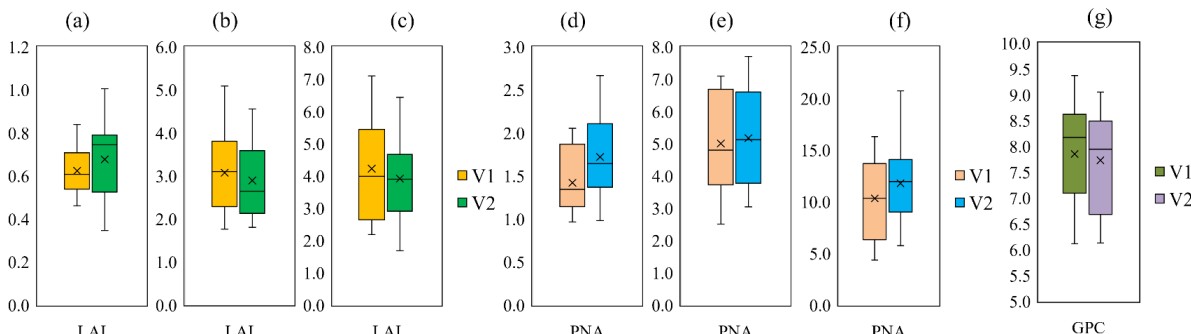

**Figure 4.** Data distribution box plots for leaf area index (LAI), plant nitrogen factor (PNA), and grain protein content (GPC) in rice varieties V1 (Meixiangzhan 2) and V2 (Wufengyou 615). LAI at (**a**) the tillering stage, (**b**) panicle initiation stage, and (**c**) heading stage; PNA at (**d**) the tillering stage, (**e**) panicle initiation stage, and (**f**) heading stage; and (**g**) GPC at the maturity stage.

*3.3. Monitoring of Factors Influencing Rice GPC*

3.3.1. Calculation of VIs in the Study Area

Three VI maps associated with crop growth status were constructed for the three growth stages in 2020 (Figure A1).

3.3.2. LAI Monitoring in the Study Area

A sampling method was used to construct spectral indices of any two-band and three-band combinations for the UAV images. Similar methods were successfully applied to relative water content (RWC) [45], LAI [46], and nitrogen [8] in wheat. Previous research shows that the best spectral parameters constructed in different forms still have collinearity, because the different forms may select the same or related sensitive bands [47]. In this study, we used multiple stepwise regression to screen the three best spectral parameters for modeling LAI and PNA. These models were evaluated in terms of $R^2$ and RMSE.

Table 3 shows the band combinations that provided the best correlations between VIs and LAI constructed using reflectance data for any two bands and any three bands after screening by the multiple stepwise regression method. All spectral indices consisting of the best bands had correlations with LAI ($r \geq 0.8$). In the panicle initiation stage, all the best spectral indices had correlations with LAI ($r \geq 0.95$). Most of the sensitive bands in the three growth stages were concentrated near 450 nm and between 670 and 730 nm, and some of them were from the red-edge region, which is consistent with the results of Tanaka et al. [46].

**Table 3.** Relationships between leaf area index (LAI) and the optimal spectral index of rice varieties V1 (Meixiangzhan 2) and V2 (Wufengyou 615) at different growth stages (*n* = 15).

| Variety | VI | Tillering Stage | | Panicle Initiation Stage | | Heading Stage | |
|---|---|---|---|---|---|---|---|
| | | Band Combination (nm) | Correlation Coefficient | Band Combination (nm) | Correlation Coefficient | Band Combination (nm) | Correlation Coefficient |
| V1 | RVI | - | - | 462,694 | 0.955 | 782,774 | 0.847 |
| | DVI | - | - | - | - | 930,934 | 0.901 |
| | IRVI | 714,718,710 | 0.880 | - | - | - | - |
| V2 | RVI | 614,502 | −0.803 | - | - | 718,722 | −0.917 |
| | NDVI | - | - | 458,674 | 0.952 | - | - |
| | PSRI | 502,602,702 | 0.890 | 458,674,474 | 0.957 | - | - |
| | SIPI | - | - | - | - | 618,722,718 | 0.954 |

The LAI regression equations for each growth stage of V1 and V2 are given in Table 4. It can be seen from Table 4 that the models based on spectral indices performed well on the modeling set (Ms) and validation set (Vs), except in the V1 tillering stage, when the $R^2$ of the validation set was only 0.55, which may have been influenced by the background noise. The rest had low RMSE values, indicating that the obtained LAI had a certain degree of confidence. Figure 5 shows scatterplots of the predicted and measured LAI values. All points of the predicted and modeled values are located on both sides of the 1:1 line. The LAI had higher estimation accuracy in the panicle initiation and heading stages than in the tillering stage. The regression equations of V1 and V2 were used to invert the LAI of the whole study area. The LAI distribution maps are shown in Figure 6.

**Table 4.** Leaf area index (LAI) regression models and their accuracy for rice varieties V1 (Meixiangzhan 2) and V2 (Wufengyou 615) at different growth stages.

| Variety | Stage | Model | Ms | | Vs | |
|---|---|---|---|---|---|---|
| | | | $R^2$ | RMSE | $R^2$ | RMSE |
| V1 | Tillering | $\text{LAI} = -184.685 \times \text{IRVI} + 93.1335$ | 0.81 | 0.05 | 0.55 | 0.04 |
| | Panicle Initiation | $\text{LAI} = 14.1343 \times \text{RVI} - 6.993$ | 0.90 | 0.30 | 0.97 | 0.16 |
| | Heading | $\text{LAI} = 100.104 \times \text{RVI} + 583.291 \times \text{DVI} - 101.89$ | 0.80 | 0.74 | 0.97 | 0.28 |
| V2 | Tillering | $\text{LAI} = 2.0792 \times \text{RVI} + 13.6268 \times \text{PSRI} + 1.767$ | 0.77 | 0.07 | 0.96 | 0.06 |
| | Panicle Initiation | $\text{LAI} = -13.348 \times \text{NDVI} + 13.0938 \times \text{PSRI} + 2.274$ | 0.95 | 0.20 | 0.92 | 0.26 |
| | Heading | $\text{LAI} = 61.34 \times \text{RVI} + 121.061 \times \text{SIPI} - 199.203$ | 0.97 | 0.32 | 0.98 | 0.48 |

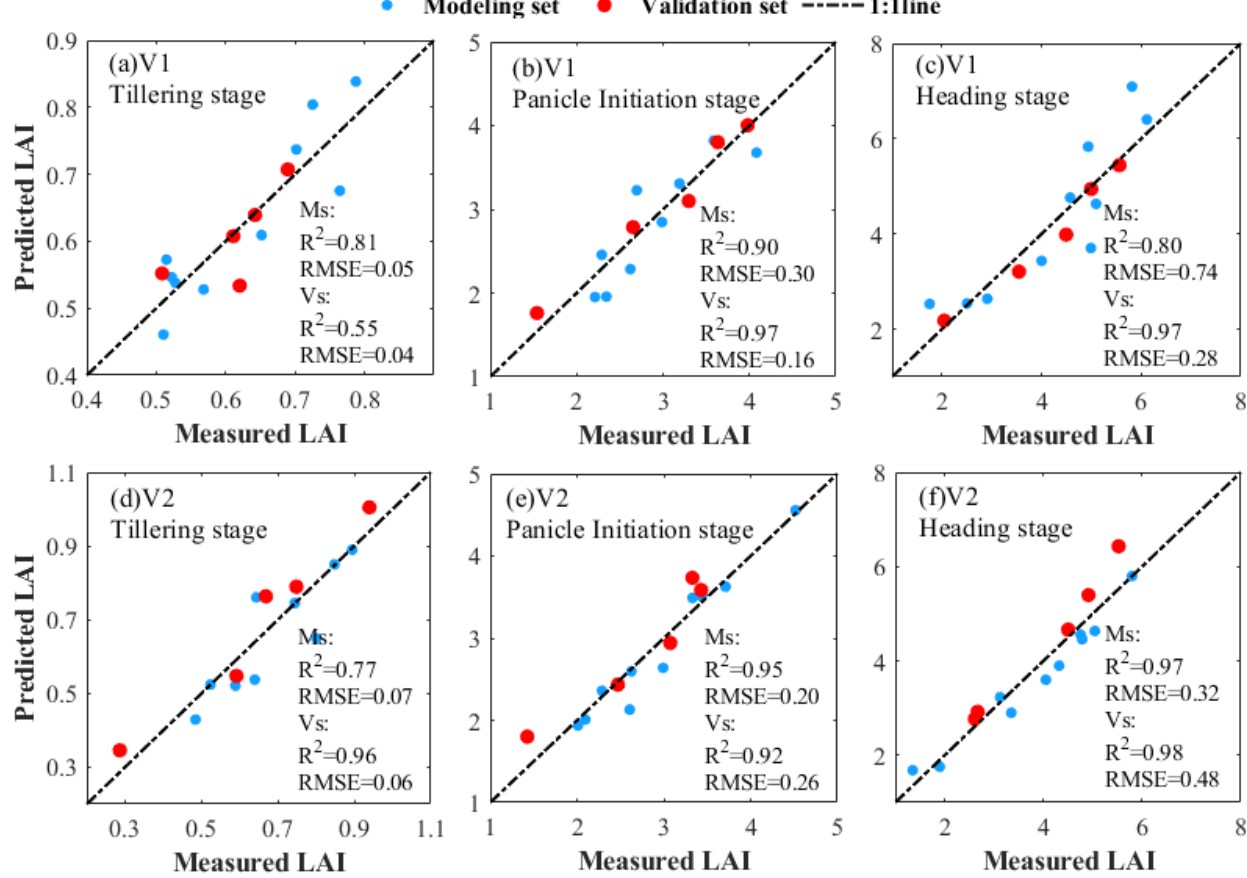

**Figure 5.** Scatterplots of measured and predicted leaf area index (LAI) values for rice varieties: V1 (Meixiangzhan 2) at (**a**) the tillering stage, (**b**) panicle initiation stage, and (**c**) heading stage; and V2 (Wufengyou 615) at (**d**) the tillering stage, (**e**) panicle initiation stage, and (**f**) heading stage.

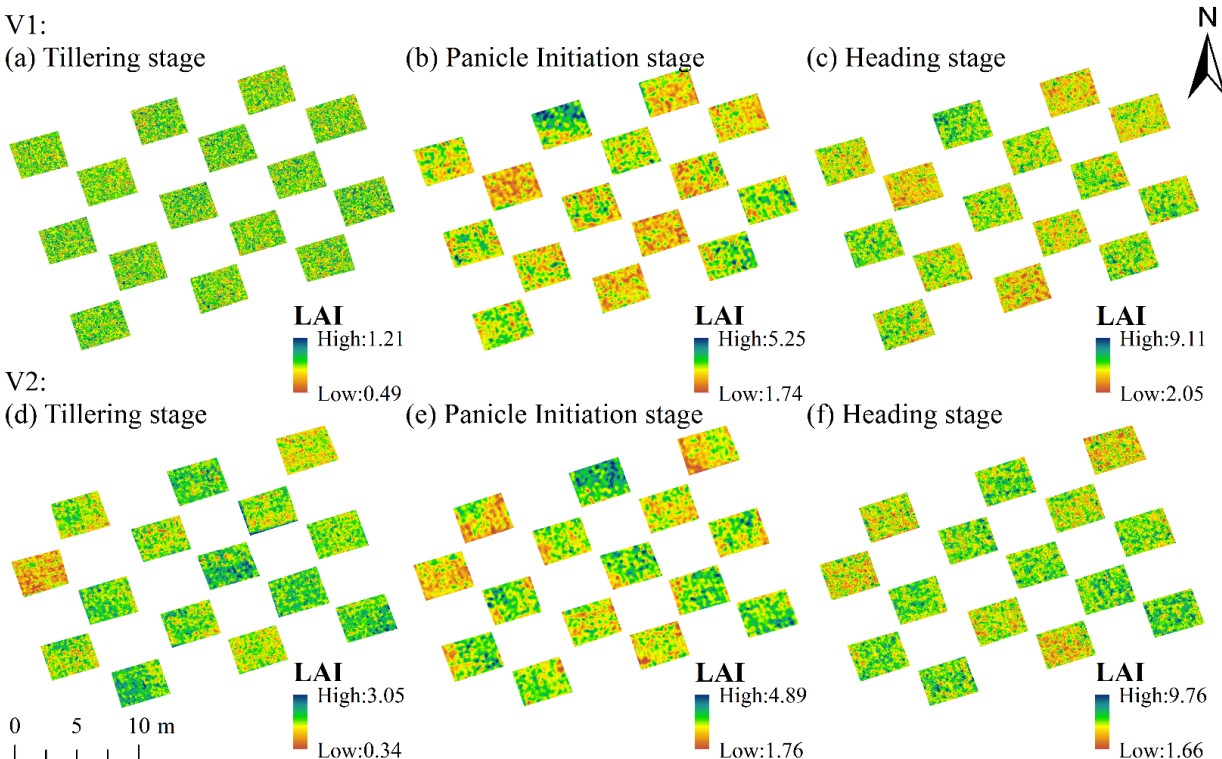

**Figure 6.** Rice leaf area index (LAI) maps of the study area. Variety V1 (Meixiangzhan 2) at (**a**) the tillering stage, (**b**) panicle initiation stage, and (**c**) heading stage; and variety V2 (Wufengyou 615) at (**d**) the tillering stage, (**e**) panicle initiation stage, and (**f**) heading stage.

### 3.3.3. PNA Monitoring in the Study Area

Nitrogen has an important role in crop growth and grain quality. Crop nitrogen accumulation (NA) is the product of crop nitrogen content (NC) and aboveground biomass (AGB). It contains information on N status and growth status [48,49] and can reflect the overall nutrient uptake of the crop well. Table 5 shows the band combinations with the best correlations between PNA and constructed VIs using UAV reflectance data selected by multiple stepwise regression methods for varieties V1 and V2. Bands more sensitive to PNA appear in the NIR region above 730 nm. It has been shown that reflectance in the NIR region is a good indicator of N status in many crops [50,51] and can indicate tissue N concentration (TNC) and biomass [52,53].

**Table 5.** Relationships between plant nitrogen accumulation (PNA) and optimal spectral indices for different rice varieties (*n* = 15).

| Variety | VI | Tillering Stage | | Panicle Initiation Stage | | Heading Stage | |
|---------|-----|------------------------------|----------------------------|------------------------------|----------------------------|------------------------------|----------------------------|
| | | Band Combination (nm) | Correlation Coefficient | Band Combination (nm) | Correlation Coefficient | Band Combination (nm) | Correlation Coefficient |
| V1 | RVI | - | - | 786,762 | 0.954 | - | - |
| | IRVI | 754,762,746 | 0.914 | 462,694,638 | 0.962 | 598,666,534 | −0.934 |
| | PSRI | 498,514,898 | 0.887 | - | - | 718,722,778 | 0.917 |
| V2 | RVI | 502,618 | 0.827 | 726,718 | 0.964 | 534,582 | 0.879 |
| | NDVI | 618,502 | −0.876 | - | - | - | - |
| | IRVI | 602,690,562 | −0.909 | 842,902,754 | 0.976 | - | - |
| | SIPI | - | - | - | - | 858,898,794 | 0.954 |

Both rice varieties V1 and V2 had good modeling and validation accuracy in all three growth stages (Table 6). Figure 7 shows scatterplots of the predicted and measured PNA values, which are more closely distributed on both sides of the 1:1 line. The $R^2$ values of the modeling and validation sets were greater than 0.86 in all three stages, except for the validation set in the tillering stage (0.79). The PNA distribution maps of the study area were obtained based on the regression equations for each growth stage (Figure 8).

**Table 6.** Plant nitrogen accumulation (PNA) regression models and their accuracy for different rice varieties ($n = 15$).

| Variety | Stage | Model | Ms | | Vs | |
|---|---|---|---|---|---|---|
| | | | $R^2$ | RMSE | $R^2$ | RMSE |
| V1 | Tillering | $PNA = 348.896 \times IRVI + 43.39 \times PSRI - 172.36$ | 0.91 | 0.11 | 0.79 | 0.13 |
| | Panicle Initiation | $PNA = 35.578 \times RVI + 28.603 \times IRVI - 41.189$ | 0.95 | 0.34 | 0.96 | 0.41 |
| | Heading | $PNA = -265.57 \times IRVI + 813.05 \times PSRI - 203.48$ | 0.88 | 0.83 | 0.99 | 0.47 |
| V2 | Tillering | $PNA = 63.159 \times RVI + 86.91 \times NDVI - 41.01 \times IRVI - 37.202$ | 0.87 | 0.15 | 0.88 | 0.22 |
| | Panicle Initiation | $PNA = 7.615 \times RVI + 96.189 \times IRVI - 58.412$ | 0.97 | 0.26 | 0.94 | 0.33 |
| | Heading | $PNA = 39.545 \times RVI + 1612.91 \times SIPI - 852.072$ | 0.90 | 1.09 | 0.99 | 0.70 |

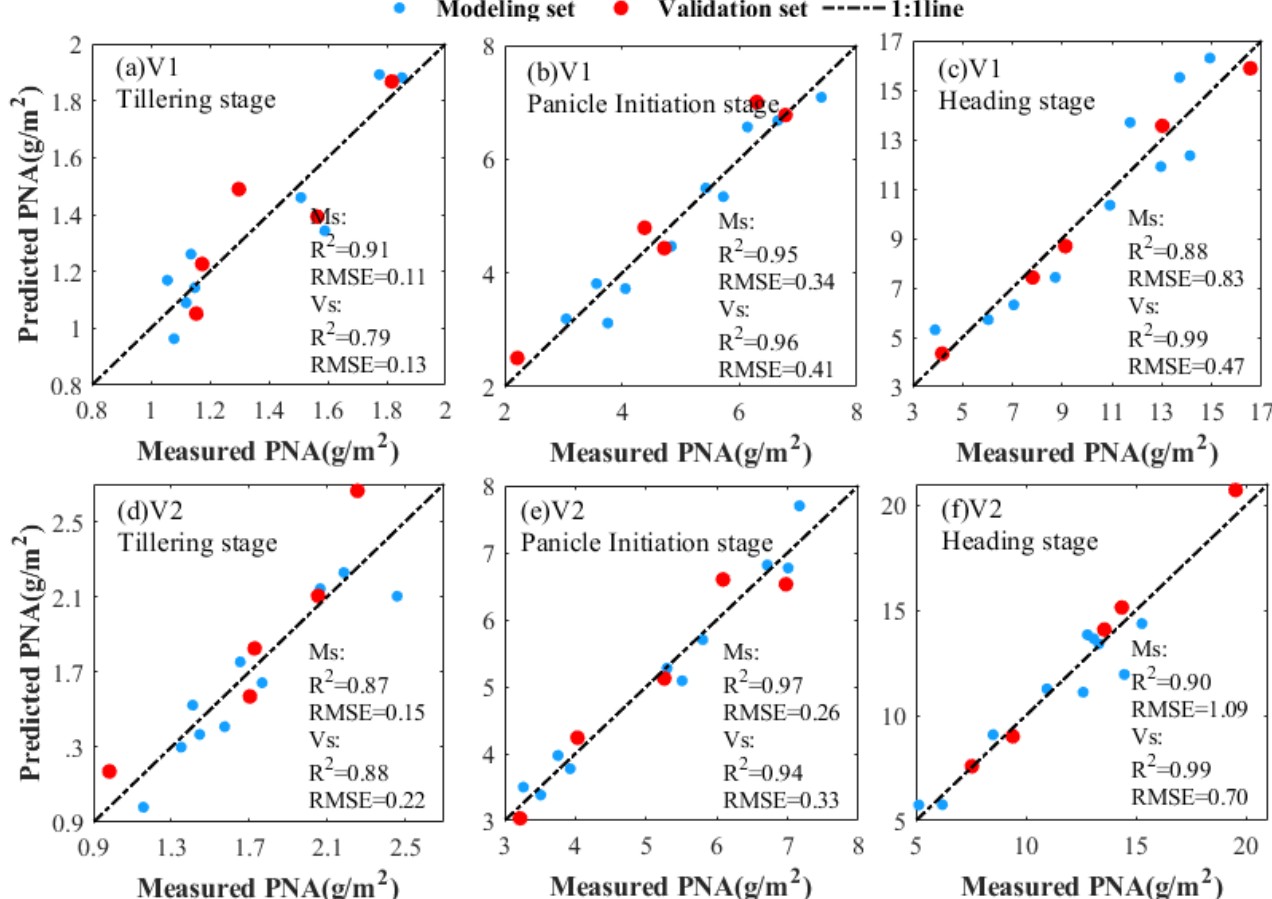

**Figure 7.** Scatterplots of measured and predicted plant nitrogen accumulation (PNA) values in variety V1 (Meixiangzhan 2) at (**a**) the tillering stage, (**b**) panicle initiation stage, and (**c**) heading stage; and in variety V2 (Wufengyou 615) at (**d**) the tillering stage, (**e**) panicle initiation stage, and (**f**) heading stage.

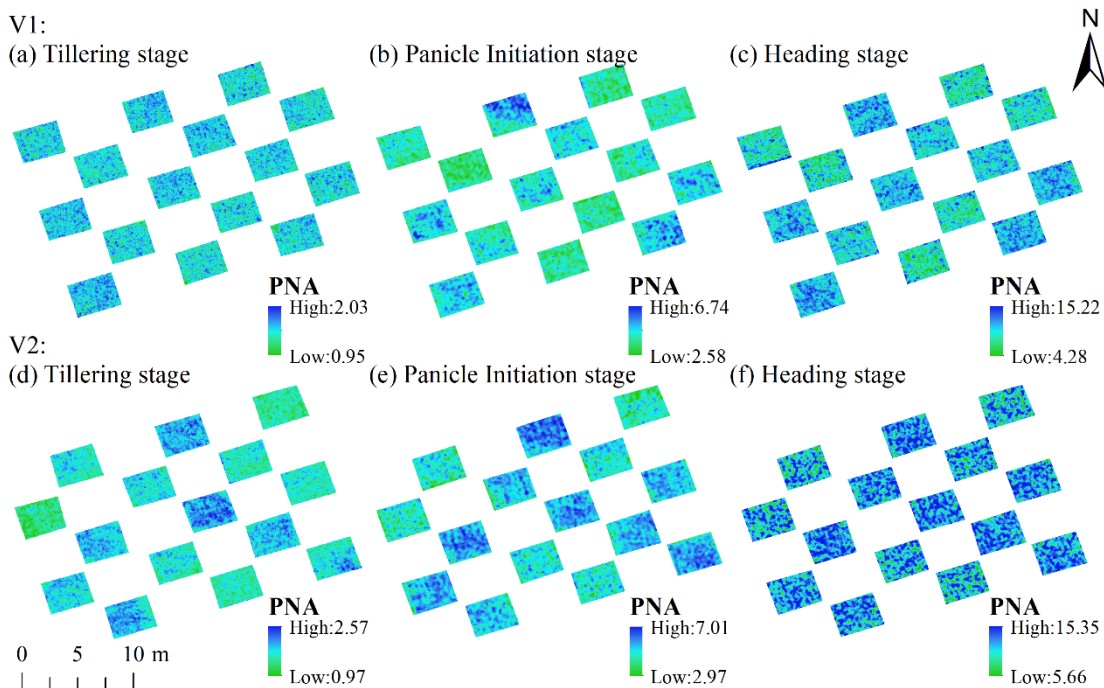

**Figure 8.** Rice plant nitrogen accumulation (PNA) distribution maps of the study area for rice variety V1 (Meixiangzhan 2) at (**a**) the tillering stage, (**b**) panicle initiation stage, and (**c**) heading stage; and for variety V2 (Wufengyou 615) at (**d**) the tillering stage, (**e**) panicle initiation stage, and (**f**) heading stage.

*3.4. GPC Monitoring Index Model Construction*

3.4.1. Correlation Analysis of Each Parameter with GPC

Table 7 shows the coefficients of correlation of rice GPC with three VIs, LAI, and PNA. Then, the best VI that was most closely related to GPC was selected and combined with LAI and PNA by the Euclidean distance method to explore their effects on GPC monitoring. From Table 7, PRI, LAI, and PNA were selected as indicators for constructing GPC monitoring indices for both varieties at the tillering stage. MTCI was selected for variety V1 and $CI_{red\ edge}$ for variety V2 to integrate LAI and PNA at the panicle initiation stage as indicators for the construction of the GPC monitoring index. PRI was selected for variety V1 and MTCI for variety V2 to incorporate LAI and PNA as indicators for constructing GPC monitoring indices at the heading stage.

**Table 7.** Correlations of grain protein content (GPC) with vegetation indices, leaf area index (LAI), and plant nitrogen accumulation (PNA) for rice varieties V1 (Meixiangzhan 2) and V2 (Wufengyou 615) at different growth stages (*n* = 15).

| Stage | Tillering Stage | | Panicle Initiation Stage | | Heading Stage | |
|---|---|---|---|---|---|---|
| **Variety** | **V1** | **V2** | **V1** | **V2** | **V1** | **V2** |
| CIred edge | 0.80 * | 0.75 | 0.89 * | 0.96 * | 0.89 * | 0.94 * |
| MTCI | 0.86 * | 0.84 * | 0.90 * | 0.94 * | 0.83 * | 0.96 * |
| PRI | 0.87 * | 0.92 * | 0.81 * | 0.93 * | 0.90 * | 0.93 * |
| LAI | 0.74 | 0.79 * | 0.76 * | 0.92 * | 0.86 * | 0.94 * |
| PNA | 0.72 | 0.80 * | 0.88 * | 0.93 * | 0.94 * | 0.91 * |

Notes: r (0.001) = 0.760, * means significance at the 0.001 level.

3.4.2. Monitoring GPC with Original Parameters

Based on the plot means of the parameters (VIs, LAI, and PNA) for V1 and V2, regression models based on each of the factors were constructed. The model equations and their accuracy are given for each growth stage of the two rice varieties in Table 8. All single-factor parameters showed some potential for GPC monitoring. Figure 9 shows the

GPC estimation map using the single factors with optimal GPC for each growth stage for varieties V1 and V2. It was found that the PRI-based GPC monitoring models for varieties V1 and V2 had slightly lower performance in the tillering stage than in the other stages, which was related to the lower rice canopy cover. The monitoring model constructed using PNA at the heading stage for variety V1 obtained optimal accuracy ($R^2$ = 0.90, RMSE = 0.29% for the modeling set and $R^2$ = 0.94 and RMSE = 0.36% for the validation set). The monitoring model for variety V2 obtained optimal accuracy using $CI_{red edge}$ at the panicle initiation stage. The $R^2$ and RMSE of the modeling set reached 0.92 and 0.26%, respectively, while the $R^2$ and RMSE of the validation set reached 0.96 and 0.16%. In addition, the $R^2$ of the monitoring model for variety V2 constructed based on MTCI at the heading stage was also greater than 0.9, showing certain monitoring stability. Nevertheless, it can be seen from the scatterplots that some of the points deviate more on both sides of the fitted line, indicating that the single-factor variables alone may not accurately express the status of the grain proteins for GPC monitoring.

**Table 8.** Rice grain protein content (GPC) estimation models based on single-factor index parameters and their accuracy (*n* = 15).

| Variety | Stage | Index | Model | Ms | | Vs | |
|---|---|---|---|---|---|---|---|
| | | | | $R^2$ | RMSE | $R^2$ | RMSE |
| V1 | Tillering | PRI | $y = 56.6x + 12.01$ | 0.74 | 0.47 | 0.78 | 0.38 |
| | | LAI | $y = -45.77x^2 + 63.38x - 13.38$ | 0.46 | 0.67 | 0.71 | 0.42 |
| | | PNA | $y = -6.74x^2 + 18.60x - 4.13$ | 0.74 | 0.46 | 0.58 | 1.27 |
| | Panicle initiation | MTCI | $y = 1.2408x + 3.6439$ | 0.84 | 0.37 | 0.77 | 0.43 |
| | | LAI | $y = -0.697x^2 + 5.47x - 1.85$ | 0.72 | 0.48 | 0.54 | 0.76 |
| | | PNA | $y = -0.256x^2 + 3.18x - 1.06$ | 0.80 | 0.40 | 0.65 | 0.67 |
| | Heading | PRI | $y = -1809.4x^2 - 88.08x + 9.98$ | 0.86 | 0.34 | 0.80 | 0.61 |
| | | LAI | $y = 0.7053x + 5.221$ | 0.76 | 0.49 | 0.73 | 0.46 |
| | | PNA | $y = 0.2152x + 5.7021$ | 0.90 | 0.29 | 0.94 | 0.36 |
| V2 | Tillering | PRI | $y = 55.53x + 11.49$ | 0.86 | 0.35 | 0.87 | 0.36 |
| | | LAI | $y = 4.23lnx + 9.42$ | 0.73 | 0.45 | 0.76 | 1.10 |
| | | PNA | $y = -1.92x^2 + 9.52x - 3.08$ | 0.76 | 0.46 | 0.56 | 1.28 |
| | Panicle initiation | $CI_{red edge}$ | $y = 2.2041x + 3.9689$ | 0.92 | 0.26 | 0.96 | 0.23 |
| | | LAI | $y = -0.32x^2 + 2.94x + 2.33$ | 0.89 | 0.30 | 0.93 | 0.47 |
| | | PNA | $y = 4.8003e^{0.1002x}$ | 0.89 | 0.33 | 0.89 | 0.29 |
| | Heading | MTCI | $y = 3.81e^{0.1878x}$ | 0.91 | 0.28 | 0.96 | 0.16 |
| | | LAI | $y = 0.7026x + 5.014$ | 0.88 | 0.33 | 0.93 | 0.25 |
| | | PNA | $y = 0.267x + 4.594$ | 0.86 | 0.36 | 0.86 | 0.32 |

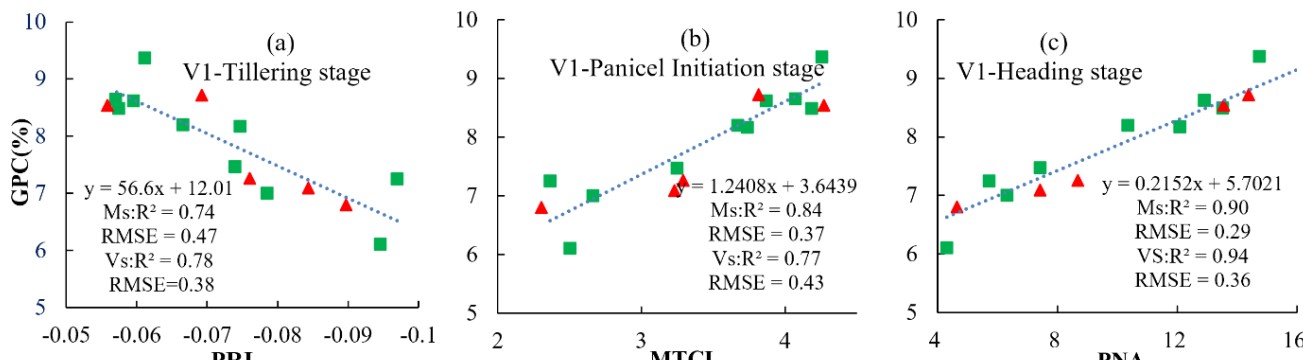

**Figure 9.** *Cont.*

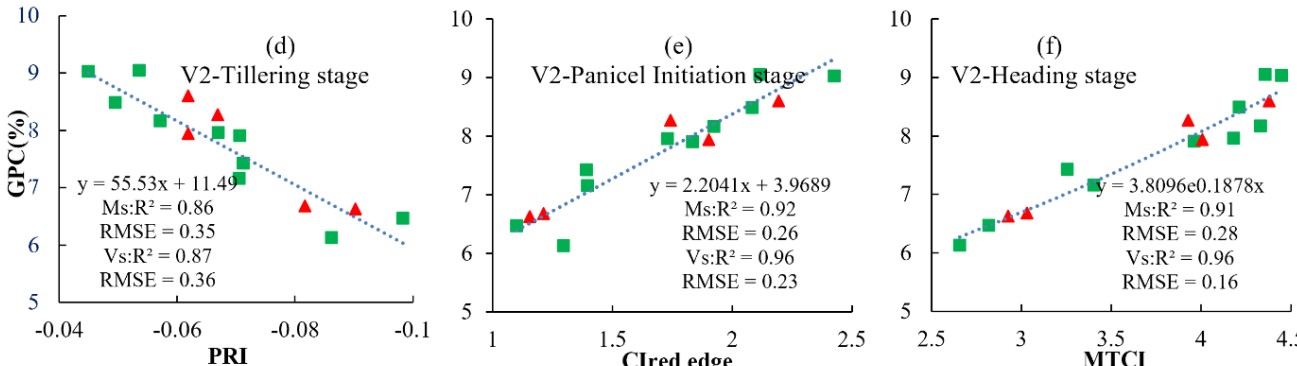

**Figure 9.** Scatterplots and regression models of grain protein content (GPC) and optimal monitoring parameters for different growth stages for variety V1 (Meixiangzhan 2) at (**a**) the tillering stage, (**b**) panicle initiation stage, and (**c**) heading stage; and for variety V2 (Wufengyou 615) at (**d**) the tillering stage, (**e**) panicle initiation stage, and (**f**) heading stage.

### 3.4.3. Two-Dimensional GPC Monitoring Index Construction and Modeling

Three Euclidean distance monitoring indices were established to monitor GPC in two-dimensional space: vegetation index–structure index, vegetation index–nitrogen index, and structure index–nitrogen index (Table 9). The constructed two-dimensional monitoring indices improved the GPC monitoring accuracy in each stage for both varieties, and the optimal monitoring accuracy $R^2$ for the modeling set of V1 at the tillering stage improved from 0.74 to 0.81 (PRI–PNA), and the accuracy of the validation set reached 0.95, and the optimal monitoring accuracy $R^2$ for the modeling set of V2 improved from 0.86 to 0.91 (PRI–LAI). The accuracy improvement in the panicle initiation and heading stages was not as great as that in the tillering stage but still improved. The GPC monitoring accuracy of variety V1 improved to 0.91 in the modeling set $R^2$ and to 0.90 in the validation set in the heading stage, and the optimal monitoring accuracy of variety V2 reached 0.94 in the modeling set $R^2$ in the panicle initiation and heading stages and reached more than 0.97 in the validation set $R^2$. Some of the GPC monitoring models had similar or even lower GPC monitoring accuracy than the original single-factor models, so it was necessary to re-evaluate and select the factors for GPC monitoring index construction by Euclidean distance, and the two-dimensional monitoring index models had more nonlinear regression equations than the single-factor models, and the larger the "distance" of these curves, the less obvious the GPC improvement effect. Figure 10 shows the scatterplots and regression models of GPC and the two-dimensional monitoring indices. Figure 10a,e reflect the nonlinear feature but, in general, the higher the index value, the higher the resulting GPC.

**Table 9.** Two-dimensional rice grain protein content (GPC) monitoring index models and their accuracy (*n* = 15).

| Variety | Stage | Index | Model | Ms | | Vs | |
|---|---|---|---|---|---|---|---|
| | | | | $R^2$ | RMSE | $R^2$ | RMSE |
| V1 | Tillering | PRI–LAI | $y = 1.664x + 6.597$ | 0.65 | 0.54 | 0.95 | 0.22 |
| | | PRI–PNA | $y = -1.264x^2 + 3.809x + 5.958$ | 0.81 | 0.40 | 0.95 | 0.38 |
| | | LAI–PNA | $y = -5.54x^2 + 9.12x - 5.116$ | 0.72 | 0.48 | 0.65 | 0.87 |
| | Panicle Initiation | MTCI–LAI | $y = 1.021 \ln x + 8.42$ | 0.84 | 0.37 | 0.55 | 0.63 |
| | | MTCI–PNA | $y = -0.214x^2 + 2.169x + 6.36$ | 0.85 | 0.34 | 0.79 | 0.44 |
| | | LAI–PNA | $y = -2.324x^2 + 5.333x - 5.771$ | 0.80 | 0.47 | 0.50 | 0.86 |
| | Heading | PRI–LAI | $y = 2.3522x + 6.137$ | 0.86 | 0.34 | 0.84 | 0.41 |
| | | PRI–PNA | $y = 2.114x + 6.238$ | 0.91 | 0.28 | 0.90 | 0.37 |
| | | LAI–PNA | $y = -0.859x^2 + 3.019x + 6.23$ | 0.88 | 0.31 | 0.81 | 0.47 |

**Table 9.** *Cont.*

| Variety | Stage | Index | Model | Ms | | Vs | |
|---|---|---|---|---|---|---|---|
| | | | | R$^2$ | RMSE | R$^2$ | RMSE |
| V2 | Tillering | PRI–LAI | $y = -0.202x^2 + 3.06x + 5.211$ | 0.91 | 0.27 | 0.83 | 0.36 |
| | | PRI–PNA | $y = -1.028x^2 + 3.93x + 5.41$ | 0.82 | 0.44 | 0.78 | 0.63 |
| | | LAI–PNA | $y = -2.420x^2 + 7.10x + 3.57$ | 0.78 | 0.40 | 0.71 | 0.49 |
| | Panicle Initiation | CI$_{red\ edge}$–LAI | $y = -0.547x^2 + 2.947x + 6.09$ | 0.94 | 0.23 | 0.98 | 0.14 |
| | | CI$_{red\ edge}$–PNA | $y = 2.103x + 6.216$ | 0.89 | 0.31 | 0.93 | 0.26 |
| | | LAI–PNA | $y = -0.811x^2 + 3.361x + 6.03$ | 0.92 | 0.28 | 0.87 | 0.22 |
| | Heading | MTCI–LAI | $y = 1.9538x + 6.185$ | 0.94 | 0.23 | 0.97 | 0.16 |
| | | MTCI–PNA | $y = 6.283e^{0.2476x}$ | 0.89 | 0.31 | 0.94 | 0.20 |
| | | LAI–PNA | $y = 0.275x^2 + 1.571x + 6.311$ | 0.89 | 0.31 | 0.92 | 0.22 |

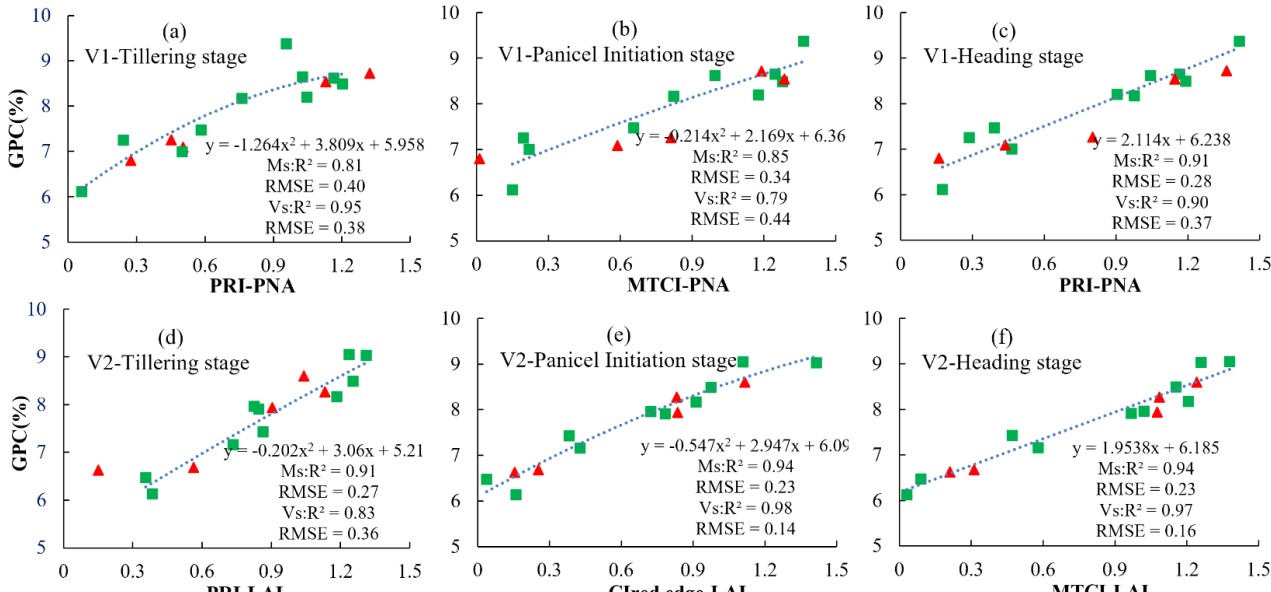

**Figure 10.** Scatterplots and regression models of grain protein content (GPC) and optimal two-dimensional monitoring indices for rice variety V1 (Meixiangzhan 2) at (**a**) the tillering stage, (**b**) panicle initiation stage, and (**c**) heading stage; and for variety V2 (Wufengyou 615) at (**d**) the tillering stage, (**e**) panicle initiation stage, and (**f**) heading stage.

3.4.4. Three-Dimensional GPC Monitoring Index Construction and Modelling

We constructed GPC monitoring indices based on VI, LAI, and PNA in three-dimensional space. Table 10 lists the three-dimensional rice GPC monitoring index models and their accuracy.

**Table 10.** Three-dimensional rice grain protein content (GPC) monitoring index models and their accuracy (*n* = 15).

| Variety | Stage | Index | Model | Ms | | Vs | |
|---|---|---|---|---|---|---|---|
| | | | | R$^2$ | RMSE | R$^2$ | RMSE |
| V1 | Tillering | PRI–LAI–PNA | $y = -1.71x^2 + 4.564x + 5.549$ | 0.76 | 0.45 | 0.86 | 0.37 |
| | Panicle Initiation | MTCI–LAI–PNA | $y = -1.05x^2 + 3.442x + 5.971$ | 0.86 | 0.35 | 0.79 | 0.44 |
| | Heading | PRI–LAI–PNA | $y = 1.878x + 6.147$ | 0.92 | 0.26 | 0.91 | 0.37 |
| V2 | Tillering | PRI–LAI–PNA | $y = -0.18x^2 + 2.833x + 5.04$ | 0.87 | 0.34 | 0.77 | 0.59 |
| | Panicle Initiation | CI$_{red\ edge}$–LAI–PNA | $y = -0.22x^2 + 2.153x + 6.12$ | 0.97 | 0.17 | 0.96 | 0.17 |
| | Heading | MTCI–LAI–PNA | $y = 6.199e^{0.22326x}$ | 0.96 | 0.20 | 0.99 | 0.15 |

Figure 11 shows the three-dimensional index models and their accuracy. The accuracy for monitoring both rice varieties at the tillering stage was no better than that with the two-dimensional monitoring index models. However, at the panicle initiation and heading stages, the three-dimensional index models were better. The modeling set $R^2$ of the three-dimensional monitoring index model constructed by PRI, LAI, and PNA at the heading stage for variety V1 reached 0.92 with an RMSE of 0.26%, and the validation set $R^2$ was 0.91 with an RMSE of 0.37%. The modeling set $R^2$ of the three-dimensional monitoring index model constructed by $CI_{red\ edge}$, LAI, and PNA at the panicle initiation stage for variety V2 reached 0.97 with an RMSE of 0.17%, and the $R^2$ of the validation set was 0.96 and the RMSE was 0.17%. The $R^2$ of both the modeling set and validation set for the three-dimensional monitoring index model constructed by MTCI, LAI, and PNA also reached above 0.96 at the heading stage. Thus, the monitoring models based on the three indices at the panicle initiation and heading stages were more accurate to monitor rice grain protein content. In Figure 11c,e,f, more points appear closely distributed on both sides of the fitted lines, achieving more reliable and stable expression for low and high levels of GPC with different optimal models for V1 and V2. Changes in grain protein synthesis by varietal factors also deserve further consideration.

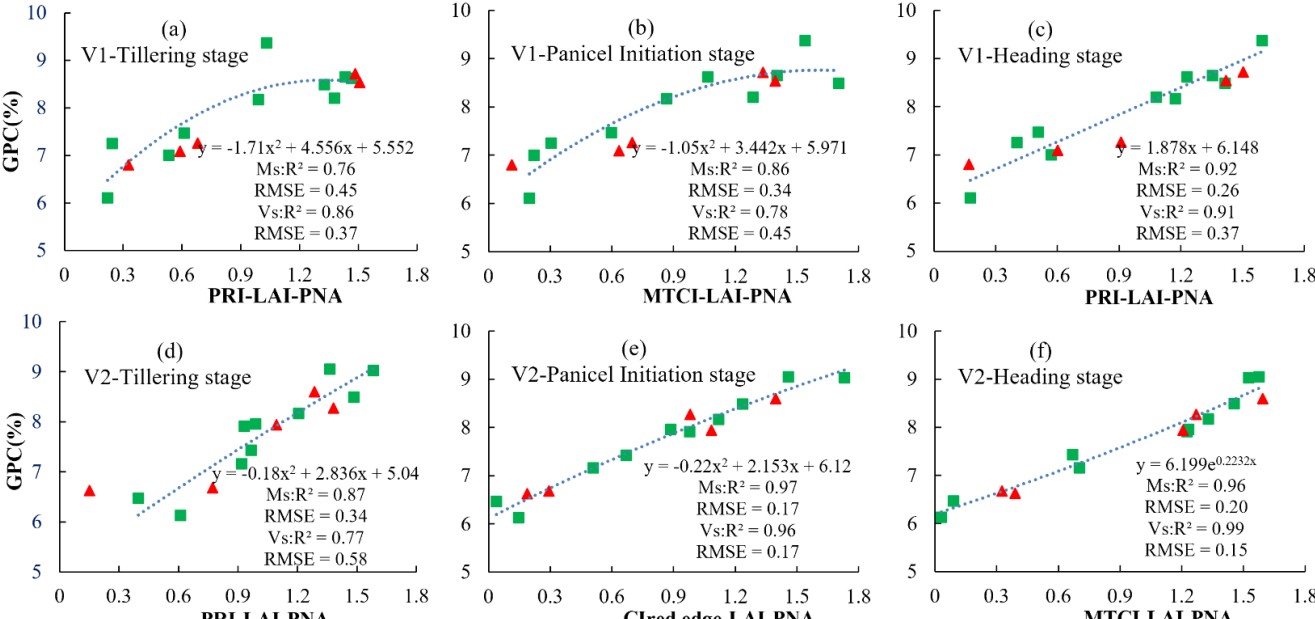

**Figure 11.** Scatterplots and regression models of grain protein content (GPC) and optimal three-dimensional monitoring indices for rice variety V1 (Meixiangzhan 2) at (**a**) the tillering stage, (**b**) panicle initiation stage, and (**c**) heading stage; and for variety V2 (Wufengyou 615) at (**d**) the tillering stage, (**e**) panicle initiation stage, and (**f**) heading stage.

### 3.4.5. GPC Monitoring Map Derived from the Optimal Models

The optimal monitoring models obtained from the three-dimensional monitoring indices for the two varieties were used for the generation of regional GPC monitoring maps. The GPC status of varieties V1 and V2 in the study area is shown in Figure 12.

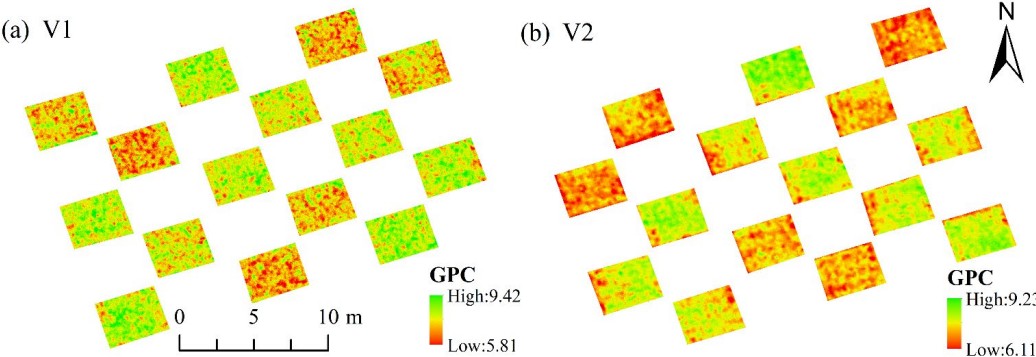

**Figure 12.** Grain protein content (GPC) monitoring maps of the study area in 2020 based on (**a**) a photochemical reflectance index (PRI)–leaf area index (LAI)–plant nitrogen accumulation (PNA) three-dimensional index model at the heading stage; and (**b**) a red edge chlorophyll index (CI$_{red\ edge}$)–LAI–PNA three-dimensional index model at the panicle initiation stage.

*3.5. Validation*

To evaluate the practical effectiveness of the models and the method, the data of V1 collected in 2019 were used to validate the LAI, PNA, and GPC models constructed by the same growth stage data of 2020. Since the UAV hyperspectral data of the study area were only acquired during the 2019 panicle initiation stage, only the models of the 2020 panicle initiation stage were used for validation.

Figure 13 shows the scatterplots of predicted and measured values for LAI and PNA in 2019 using the LAI and PNA sensitive spectral parameters and monitoring models for V1 in 2020. Compared to the accuracy in 2020, the monitoring accuracy in 2019 was somewhat lower in all cases, with $R^2 = 0.77$ and RMSE = 0.26% for the LAI modeling set, $R^2 = 0.85$ and RMSE = 0.40% for its validation set, $R^2 = 0.78$ and RMSE = 0.71% for the PNA modeling set, and $R^2 = 0.68$ and RMSE = 0.16% for its validation set. Although the same rice varieties and cultivation management practices were used in the two years, there were differences in macroclimatic factors and moisture that may have affected rice growth. The scatterplots in Figure 13 show that the predicted LAI and PNA in 2019 based on the 2020 models still have some reliability and can be used for the generation of regional LAI and PNA maps (Figure 14) for GPC monitoring.

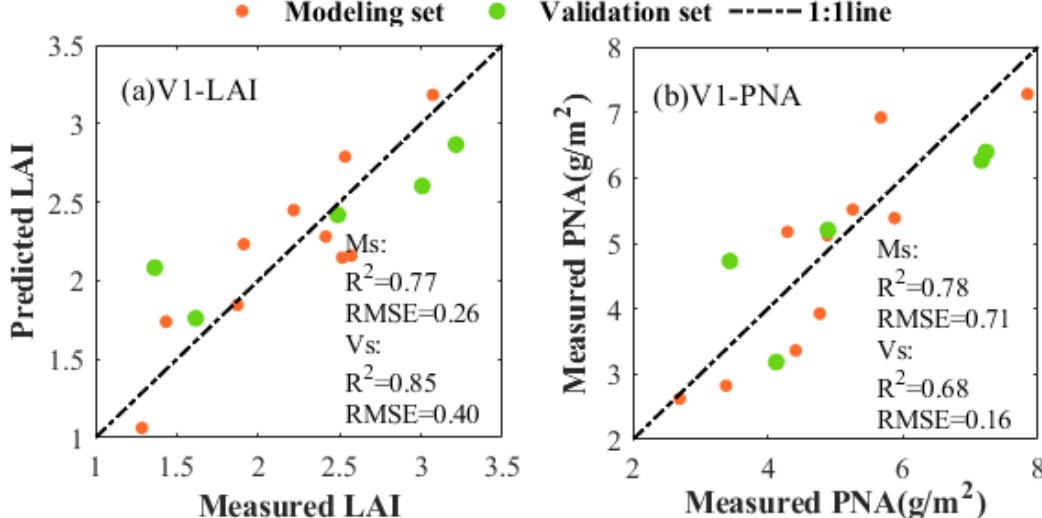

**Figure 13.** Scatterplots of measured and predicted (**a**) leaf area index (LAI) and (**b**) plant nitrogen accumulation (PNA) in 2019.

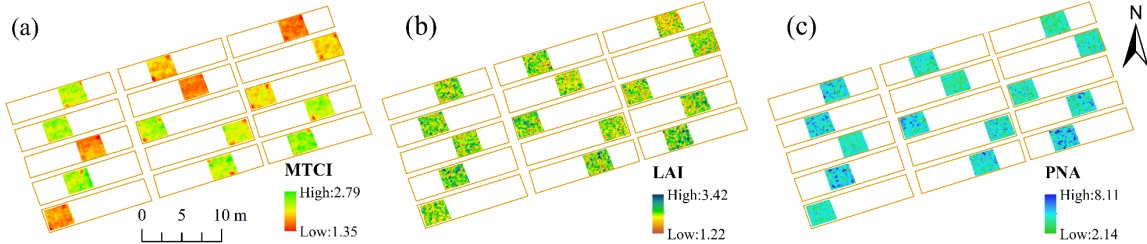

**Figure 14.** Monitoring maps for rice variety V1 (Meixiangzhan 2) in the panicle initiation stage in 2019 for the study area: (**a**) MERIS terrestrial chlorophyll index (MTCI), (**b**) leaf area index (LAI), and (**c**) plant nitrogen accumulation (PNA).

By calculating the mean values of MTCI, LAI, and PNA for the 15 nitrogen plots in 2019, a GPC model based on MTCI–LAI–PNA data was obtained by applying the three-dimensional Euclidean distance method (Figure 15). The $R^2$ and RMSE were 0.70 and 0.69%, respectively, for the modeling set and 0.77 and 0.84% for the validation set, respectively, indicating that the model has a certain adaptability in time. However, the accuracy was still lower than that of the modeling set ($R^2 = 0.86$, RMSE = 0.35%) and validation set ($R^2 = 0.79$, RMSE = 0.44%) of the three-dimensional monitoring index model for the panicle initiation stage in 2020. In addition, the regional GPC monitoring map for 2019 was obtained in this study based on the uncorrected model for 2020 (Figure 16).

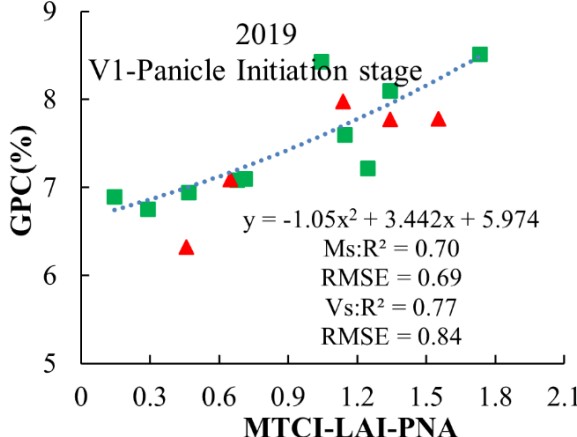

**Figure 15.** Grain protein content (GPC) regression model based on the three-dimensional monitoring index of rice variety V1 (Meixiangzhan 2) for the panicle initiation stage in 2019.

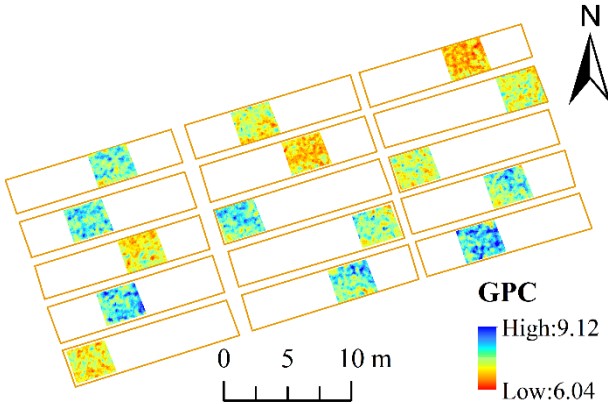

**Figure 16.** Grain protein content (GPC) map of rice variety V1 (Meixiangzhan 2) in 2019 based on the 2020 three-dimensional monitoring index model.

## 4. Discussion

### 4.1. LAI and PNA Monitoring Based on Sensitive Spectral Indices

LAI is an important parameter describing the structure of the crop canopy. It determines, to some extent, the photosynthesis, transpiration, carbon and nitrogen cycling, and water vapor interception [31,32]. Accurate monitoring of LAI and PNA in all stages of rice growth can provide an accurate description of crop growth conditions. In this study, the sensitive spectral indices of LAI and PNA of different rice varieties at different growth stages were screened by constructing spectral indices of arbitrary band combinations. The use of LAI and PNA estimated from spectral information for subsequent rice GPC monitoring is an attempt to simplify the data acquisition process, minimizing manual sampling and laboratory assays to obtain LAI and PNA. On the other hand, these sensitive spectra were not completely consistent across the varieties and growth stages, as there were changes in the responses to spectra at different growth stages. Finding a new and reliable spectral index is a key issue in agricultural remote sensing monitoring [48,54]. In addition to obtaining sensitive bands, like previous studies, this study also found that some spectral combinations were not common spectral indices used to estimate LAI and PNA, such as the sensitive spectral parameter DVI (930,934) for LAI of the variety V2 at the heading stage. The sensitive band appeared after 900 nm, although this NIR band was considered undesirable in previous studies due to high collinearity [47,55]. In the panicle initiation and heading stages, the models show better accuracy using both the modeling and validation sets because they are less affected by background information. In the tillering stage, the background information is still the main reason for the poor performance of the monitoring model [56].

### 4.2. Rice GPC Monitoring Based on Euclidean Distance

The Euclidean distance-based modeling approach combines the factors that influence rice quality into a numerical metric that allows the effects of multiple factors on rice grain protein synthesis to be quantified.

To fully consider the influence of multidimensional factors on rice GPC, we constructed a comprehensive GPC monitoring index that considers the crop's phenological state, phenotypic structural characteristics, and nutrient status. Although the different VIs express different vegetation states, there is a certain amount of multicollinearity due to the duplication of information existing between spectral bands. In order to remove the influence of covariance, multiple stepwise regression method was used to select parameters carrying as much GPC-related information as possible in this study. It is necessary to compare the methods proposed in this study with those that are already more mature and commonly used methods. Tables A1 and A2 list the rice GPC estimation results through two-factor and three-factor multiple linear regression models and their accuracy. Table A3 shows the relationship between the GPC and optimal spectral indices for different varieties, while Table A4 lists the GPC estimation results using the regression method with arbitrary combinations of spectral indices and their accuracy. By comparison, the Euclidean distance method ensured better monitoring accuracy at all growth stages.

By calculating the Euclidean distance [42], we can know that a negative factor in rice growth may inhibit the facilitation effect of positive factors, while multiple positive factors will maintain or enhance this facilitation effect, which is more in line with the natural state of the rice grain protein synthesis process [11]. The GPC monitoring index based on Euclidean distance was applied to two rice varieties in three growth stages (Figures 10a,d and 11b,c,e,f). A multi-factor index combined by Euclidean distance can provide better GPC prediction accuracy than a single-factor index. GPC monitoring models based on different rice varieties can avoid monitoring errors caused by variety effects. This study reveals that the factors affecting GPC differ in each stage for different rice varieties, which is consistent with the findings of Devi et al. [57]. For better estimation accuracy, the above factors should also be considered when the model is migrated. The model can also explain the crop growth difference caused by stage change for the two rice

varieties and provide a reference for establishing a GPC monitoring model based on crop variety information.

In this study, we established rice GPC estimation models from crop VIs, LAI, and PNA based on two years of experiments without considering the variation in climatic and other uncontrollable conditions. Temperature is widely considered to be an important driver of crop growth [58,59], so it must be considered when the model is extended to the regional scale. The proposed method effectively estimated rice GPC using UAV images. Previous studies also attempted to monitor crop growth and grain quality through satellite imagery [60,61]. The Euclidean distance method has a simpler data processing process and has the potential to be applied to satellite data.

### 4.3. Potential of Applying UAV-Obtained Hyperspectral Data to Crop Monitoring

In this study, two important rice growth parameters, LAI and PNA, were estimated through the VIs collected from the UVA images. Then, a complete GPC monitoring model combining two- or three-dimensional crop growth parameters was developed using hyperspectral data obtained by UAV. The results from this study indicate that UAV hyperspectral image data have good potential on cereal crop quality monitoring [62,63]. However, GPC sensitive parameters may differ among crop varieties and regions, so more research is needed in future applications of this method to GPC monitoring [25]. This study demonstrates a more objective approach for observation and visualization of actual rice GPC distributions of different varieties and provides a UAV-based remote sensing tool that has the potential for rapid crop monitoring on a regional scale.

### 5. Conclusions

In this study, rice GPC monitoring models were developed for different rice varieties at critical growth stages using the Euclidean distance method. The following conclusions were obtained:

(1) The spectral indices based on arbitrary band combinations can effectively monitor LAI and PNA in rice at different growth stages. The estimation accuracy ($R^2$) for two rice varieties in three growth stages exceeded 0.8. These spectral indices can be used to generate LAI and PNA distribution maps.

(2) The two-dimensional GPC monitoring index model based on PRI and PNA provided the optimal monitoring accuracy for variety V1 at the tillering stage, with $R^2$ and RMSE of 0.81 and 0.40%, respectively, for the modeling set and 0.95 and 0.38%, respectively, for the validation set. The optimal two-dimensional index model based on PRI and LAI for variety V2 at the tillering stage had $R^2$ and RMSE of 0.91 and 0.27% for the modeling set and 0.83 and 0.36% for the validation set. The three-dimensional GPC monitoring index model based on MTCI, LAI, and PNA provided the optimal monitoring accuracy for variety V1 at the panicle initiation stage, with the modeling $R^2$ and RMSE being 0.86 and 0.35% and the validation $R^2$ and RMSE being 0.79 and 0.44%. The $R^2$ and RMSE for the optimal three-dimensional model based on $CI_{red\ edge}$, LAI, and PNA for variety V2 at the initiation stage were 0.97 and 0.17% for the modeling set and 0.96 and 0.17 for the validation set. The three-dimensional GPC monitoring index model based on PRI, LAI, and PNA provided the optimal monitoring accuracy for variety V1 at the heading stage, with $R^2$ and RMSE of 0.92 and 0.26% for modeling and 0.91 and 0.37% for validation. The optimal three-dimensional model based on MTCI, LAI, and PNA for variety V2 at the heading stage had $R^2$ and RMSE of 0.96 and 0.20% for modeling and 0.99 and 0.15% for validation. In addition, single-factor indices had linear relationships with GPC, whereas most of the composite GPC monitoring indices had nonlinear relationships with GPC.

(3) The models and methods from this study have the potential for use in the field of UAV-based remote sensing for crop monitoring. It is feasible to adapt the models for GPC monitoring on the same rice varieties in different years. In this study, a limited number of key influencing factors that affect rice quality and yield were considered. The effects of a wide range of influencing factors and growth stages on the construction of GPC need to be

further investigated. The applicability of the model to different rice varieties also needs to be further validated.

**Author Contributions:** J.Z. and X.S. processed and analyzed the data and drafted the manuscript. X.S. guided the experimental design, participated in data collection, advised on data analysis, and revised the manuscript. X.J., G.Y., C.Y., H.F., J.W. and S.M. were involved in the experiments, ground data collection, and/or manuscript revision. All authors read and approved the final version. All authors have read and agreed to the published version of the manuscript.

**Funding:** This research was funded by Key-Area Research and Development Program of Guangdong Province (2019B020214002) and National Natural Science Foundation of China (42171394).

**Acknowledgments:** We appreciate the help from Youqiang Fu, Hong Chang and Weiguo Li during field data collection.

**Conflicts of Interest:** The authors declare no conflict of interest.

**Appendix A**

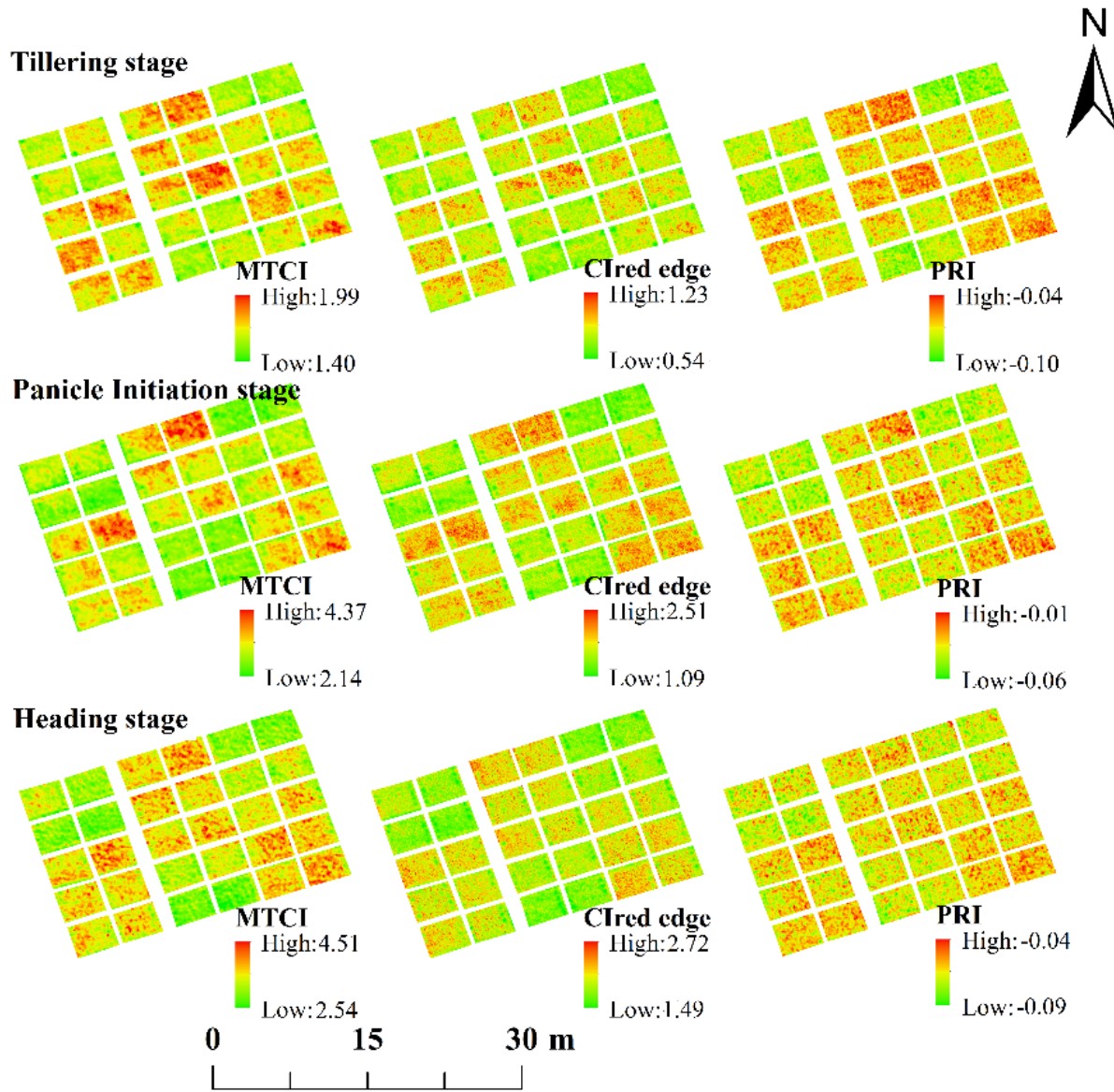

**Figure A1.** Rice vegetation index monitoring maps of the experimental area at different growth stages.

**Table A1.** Two-factor multiple linear regression rice grain protein content (GPC) estimation models and their accuracy (*n* = 15).

| Variety | Stage | Index | Model | Modeling Set | | Validation Set | |
|---|---|---|---|---|---|---|---|
| | | | | $R^2$ | RMSE | $R^2$ | RMSE |
| V1 | Tillering | PRI–LAI | $y = 50.31PRI + 1.56LAI + 10.58$ | 0.61 | 0.57 | 0.72 | 0.48 |
| | | PRI–PNA | $y = 53.16PRI + 0.25PNA + 11.37$ | 0.73 | 0.48 | 0.83 | 0.47 |
| | | LAI–PNA | $y = 6.96LAI + 0.14PNA + 3.04$ | 0.41 | 0.85 | 0.67 | 0.74 |
| | Panicle Initiation | MTCI–LAI | $y = 0.35MTCI + 0.61LAI + 5.21$ | 0.71 | 0.48 | 0.67 | 0.53 |
| | | MTCI–PNA | $y = 0.39MTCI + 0.58PNA + 4.11$ | 0.78 | 0.41 | 0.78 | 0.42 |
| | | LAI–PNA | $y = -0.26LAI + 0.62PNA + 4.10$ | 0.69 | 0.53 | 0.73 | 0.47 |
| | Heading | PRI–LAI | $y = 82.15PRI + 0.29LAI + 12.15$ | 0.80 | 0.37 | 0.83 | 0.35 |
| | | PRI–PNA | $y = 81.2PRI + 0.08PNA + 12.39$ | 0.86 | 0.36 | 0.89 | 0.37 |
| | | LAI–PNA | $y = 0.34LAI + 0.11PNA + 5.42$ | 0.75 | 0.46 | 0.91 | 0.42 |
| V2 | Tillering | PRI–LAI | $y = 53.30PRI + 0.38LAI + 11.14$ | 0.86 | 0.36 | 0.87 | 0.35 |
| | | PRI–PNA | $y = 55.38PRI + 0.08PNA + 11.38$ | 0.81 | 0.41 | 0.79 | 0.45 |
| | | LAI–PNA | $y = 3.97LAI + 0.18PNA + 4.80$ | 0.72 | 0.56 | 0.64 | 0.59 |
| | Panicle Initiation | $CI_{red\ edge}$–LAI | $y = 0.76CI_{red\ edge} + 1.27LAI + 2.46$ | 0.86 | 0.37 | 0.87 | 0.32 |
| | | $CI_{red\ edge}$–PNA | $y = 0.93CI_{red\ edge} + 0.78PNA + 1.86$ | 0.86 | 0.38 | 0.95 | 0.21 |
| | | LAI–PNA | $y = 0.56LAI + 0.28PNA + 4.82$ | 0.87 | 0.33 | 0.86 | 0.33 |
| | Heading | MTCI–LAI | $y = 1.34MTCI + 0.14LAI$ | 0.88 | 0.32 | 0.89 | 0.35 |
| | | MTCI–PNA | $y = 1.40MTCI + 0.10PNA + 2.19$ | 0.91 | 0.27 | 0.91 | 0.26 |
| | | LAI–PNA | $y = 0.65LAI + 0.02PNA + 4.93$ | 0.87 | 0.33 | 0.79 | 0.40 |

**Table A2.** Three-factor multiple linear regression rice grain protein content (GPC) estimation models and their accuracy (*n* = 15).

| Variety | Stage | Index | Model | Modeling Set | | Validation Set | |
|---|---|---|---|---|---|---|---|
| | | | | $R^2$ | RMSE | $R^2$ | RMSE |
| V1 | Tillering | PRI–LAI–PNA | $y = 49.72PRI + 1.19LAI + 0.13PNA + 10.58$ | 0.65 | 0.52 | 0.79 | 0.43 |
| | Panicle Initiation | MTCI–LAI–PNA | $y = 0.12MTCI - 0.19LAI + 0.57PNA + 5.41$ | 0.77 | 0.45 | 0.91 | 0.31 |
| | Heading | PRI–LAI–PNA | $y = 72.39PRI + 0.21LAI + 0.04PNA + 11.41$ | 0.87 | 0.34 | 0.87 | 0.34 |
| V2 | Tillering | PRI–LAI–PNA | $y = 53.21PRI + 0.32LAI + 0.03PNA + 11.12$ | 0.85 | 0.36 | 0.87 | 0.32 |
| | Panicle Initiation | $CI_{red\ edge}$–LAI–PNA | $y = 0.04CI_{red\ edge} + 0.58LAI + 0.27PNA + 4.82$ | 0.88 | 0.33 | 0.86 | 0.31 |
| | Heading | MTCI–LAI–PNA | $y = 0.41MTCI - 0.08LAI + 0.19PNA + 4.64$ | 0.77 | 0.45 | 0.87 | 0.31 |

**Table A3.** Relationships between grain protein content (GPC) and optimal spectral indices for two rice varieties at three different growth stages (*n* = 15).

| Variety | VIS | Tillering Stage | | Panicle Initiation Stage | | Heading Stage | |
|---|---|---|---|---|---|---|---|
| | | Band Combination (nm) | Correlation Coefficient | Band Combination (nm) | Correlation Coefficient | Band Combination (nm) | Correlation Coefficient |
| V1 | RVI | 528,702 | 0.732 | 774,782 | −0.828 | 782,774 | 0.816 |
| | DVI | 554,706 | −0.836 | 782,774 | 0.835 | - | - |
| | SIPI | - | - | - | - | 706,722,782 | 0.923 |
| V2 | NDVI | - | - | 726,730 | −0.866 | - | - |
| | PSRI | 462,464,770 | 0.901 | 606,586,778 | 0.870 | 766,758,878 | 0.881 |
| | SIPI | - | - | - | - | 758,766,930 | 0.881 |

**Table A4.** Grain protein content (GPC) regression models and their accuracy for two rice varieties at three different growth stages ($n = 15$).

| Variety | Stage | Model | Modeling set | | Validation set | |
|---------|-------|-------|:---:|:---:|:---:|:---:|
| | | | $R^2$ | RMSE | $R^2$ | RMSE |
| V1 | Tillering | $GPC = -12.035 \times RVI + 11.525 \times DVI + 26.092$ | 0.69 | 0.54 | 0.79 | 0.50 |
| | Panicle Initiation | $GPC = 143.534 \times RVI + 667.438 \times DVI - 137.007$ | 0.81 | 0.37 | 0.83 | 0.36 |
| | Heading | $GPC = 173.728 \times RVI + 0.871 \times SIPI - 63.49$ | 0.84 | 0.36 | 0.79 | 0.48 |
| V2 | Tillering | $GPC = 8.476 \times RVI + 15.464$ | 0.81 | 0.37 | 0.88 | 0.29 |
| | Panicle Initiation | $GPC = -49.88 \times NDVI + 145.865 \times PSRI + 5.819$ | 0.89 | 0.29 | 0.92 | 0.28 |
| | Heading | $GPC = 39.545 \times RVI + 1612.91 \times SIPI - 852.072$ | 0.90 | 0.27 | 0.90 | 0.26 |

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
