# Peer review of "Remote Sensing Monitoring of Rice Grain Protein Content Based on a Multidimensional Euclidean Distance Method"

_remotesensing, doi:10.3390/rs14163989_

Round 1

Reviewer 1 Report

This is an exceptionally well-done study and I recommend its publication with only a couple of minor comments:

1. Section 2.2.2: Can you provide an intuitive explanation for the equation to calculate LAI?

2. In the discussion section, it would be helpful to show how your approach will potentially work if you were to use Landsat or Sentinel satellite data.

Author Response

Thank you for your review. We have answered your comments in the document "Response to Review Comments", please review it.

Reviewer 2 Report

Review to Remotesensing-1758582

The grain protein content (GPC) of rice is an important factor affecting its nutrition and taste. Monitoring crop nutrition status is essential for guiding the fertilizer management and improving cereal quality. This study developed rice GPC monitoring models for different rice varieties at critical growth stages using Euclidean distance method. However, such method does not essentially achieve the authors’ original intention of combining multiple affecting factors. Thus, this paper is not appropriate to publish.

Major comments:

1.     It’s a good idea to consider multiple factors in one explicit and practical index. However, the indices that construct multidimensional space exist a certain amount of collinearity, as the LAI and PNA is also estimated from spectral information. From my point of view, the proposed model does not essentially combine influenced factors from different aspects, but still reveal the relationship between GPC and spectral information like previous studies.

2.     The accuracy improvement by using Euclidean distance method is limited. The accuracy of rice GPC estimation models based on original single parameter are high and acceptable (Table 8), especially for the monitoring model for variety V2 using CIred edge at panicle initiation stage, the R2 can achieve 0.92. When adding additional LAI, the accuracy only improve 0.02 (Table 9). From the perspective of practical application and model simplicity, only CIred edge is enough.

3.     The advantage of Euclidean distance method should be emphasized. In my understanding, the application of the Euclidean distance is to reduce dimension so that, two-dimensional or three-dimensional indices could be converted to a single value, namely, the distance from the point in the two or three-dimensional space to the origin. But how does this method maximize the information carried in the indices? There are other ways to reduce dimension, or to create multivariate index, why this approach? Hs this method successfully be applied in other fields? Is there any practical meaning for the distance? Detailed description is necessary.

4.     If the study is still willing to conduct more detailed analysis of proposed model, considering analyze the contribution of different factors as such information may be already contained in the distance. If the PRI, LAI, and PNA are used to predict GPC, why don’t the authors run multiple linear regression or machine learning algorithm to derive an empirical relationship? 

5.     Although the experiment is designed with five nitrogen fertilization levels, the accuracy evaluation is based on global treatment. Different fertilization management may result in variations in model performance.

Specific comments:

1.     Should “PNC” be modified to “PNA” is Line 101?

2.     It would be better to express the practical meaning of Euclidean distances with colors in schematic diagram (Figure 2). The content in this figure should be richer.

3.     It is easier for readers to discover the improvement when the accuracy from single to multiple factors is placed in one table.

Author Response

(The authors gave the same response as above.)

Reviewer 3 Report

The article entitled Remote Sensing Monitoring of Rice Grain Protein Content Based on a Multidimensional Euclidean Distance Method is an article of public interest, I read the paper with interest.

However, there are some aspects that can be improved

the summary and introduction are well sized compared to the rest of the article.

I think it would be more representative for the plant development stage if BBCH-scale were used, I think this would improve the manuscript. also the number of cited articles could be improved and recent articles added.

I think that in order to better capitalize on the results obtained, figures 5,7, 9, 13 and 15 can be moved to a Supplementary Files for a better image quality, and the free space obtained can be capitalized by resizing the images 6,8, 10 , 11 and 12, and so probably the title of figure 12 will be on the same page as figure.

Otherwise the results are interesting and clearly presented.

Author Response

(The authors gave the same response as above.)

Reviewer 4 Report

First of all, congratulation on the good work. It is a shame that data was not acquired for all 3 stages in 2019, it would have made your paper great. Nonetheless, the way the monitoring factors are integrated is very interesting. I have some comments that I hope will improve your article, and I will try to list them from the most important to the least.

In lines 89 and 90, where the 3 types of factors that are considered for GPC estimation are listed, there needs to be a discussion on why the authors chose exactly those categories. For example, why not include a water-related factor?

Similarly, in lines 95 and 96, where the specific factors are listed for each category, there needs to be discussion on why those were the chosen ones. For example, why was LAI chosen as the canopy structure factor instead of others like canopy rugosity or clumping index?

In Section 2.3.1, there needs to be more discussion on why only those 3 indices were considered. In particular, why not use the arbitrary band combination for the GPC directly? I understand the results might not be as good as the combination with LAI and PNA, but it still surprises me not to see it tried.

In Section 2.2.2, citations should be added for the LAI and GPC determination procedures.

In line 152, it would be good to have the equations used to convert to reflectance values.

In lines 76 to 78, citations need to be added to support the claims made.

In the second conclusion, the use of the word "optimal" might imply that it is the best possible accuracy, when I suspect the authors just meant the best out of the evaluated models. Changing the word or rephrasing the sentences where it is used would be better.

In line 47, where [15] is cited, it does not seem to support appropriately the preceding sentence.

In line 395, the acronym AGB needs to be defined.

In line 49, the sentence where [2,3] are cited, would benefit from having instead citations that deal with rice too.

In lines 79 and 80, why would UAV data be considered a bridge between ground-based and satellite remote sensing? It would need either to be rephrased or to add a supporting citation.

In Section 2.4, the discussion about Euclidean distance can be simplified. It is effectively just a method to combine the scores from each factor into an overall score.

In line 117, planting density for 2020 should be reported with the same style as that used for 2019 in line 113.

In line 47, the sentence "...traded in a higher prices than normal rice in market." needs to be reworded.

In line 576, the word "extent" is repeated back to back.

Author Response

(The authors gave the same response as above.)

Reviewer 5 Report

This paper needs major revision.

Author Response

(The authors gave the same response as above.)

Reviewer 6 Report

General comments:

The study is well planned and detailed in the structure of the tests. The selection of two varieties, the possibility of using a hyperspectral camera, the number of plots used and the typologies of the N0 - N4 treatments performed, express a wide opportunity to validate the research conducted. The study of three crop phases is also interesting and yet only one of the crop phases (growth) is available in 2019 to compare the data in two years of realization and to be able to conclude in a more important way. This would have been a great opportunity that could not have been considered, but it does not detract from the article because of the wide range of tests performed.

 For these reasons, the reviewer considers that the work has a sufficient level of quality to be published.

However, in the reviewer's opinion, different improvements should be made to it in order to get it published:

Specific aspects:

1.- The explanation of the data used should be improved:

- In line 114 it is indicated that 15 plots are taken into account for the 2019 study, but it is not indicated how many are used in 2020, perhaps all?

- In 2020 it is indicated that the manual planting is 20 x 20cm and in 2019 1 x 20cm. Is this correct? It is indicated below that management is the same, can you clarify this????

- In the case of studies carried out with UAV, it is clear that the height of the flights is 30m, but it is not clearly indicated which is the GSD of the flight unless it is the 80 x 80 cm of line 156 but it is not well expressed.

- In line 353 there is a clear reference to 2020 data. I think the same should be done in 3.1 and 3.2.

- In 212 - 215 the indexes used are explained, but IRVI is left out, it should be included in the explanation.

- Line 199 see UVA and should be UAV.

2.- Regarding the figures:

- Figures should be brought closer to the epigraph where they are commented because they make the reader have to scroll down at once to understand the comments made by the researchers.

- Specifically:

o On line 111 it could be indicated that Fig 1a refers to 2019.

o On line 115 the same could be done with Fig 1b in 2020.

o This would correct that the reference to Fig 1 appears on line 136 when the figure itself is on line 120.

- Figure 3 should be moved up a bit to be able to follow the explanation of its composition with respect to the graph.

- In figure 6 and 8 it would be more conclusive to put the regression line drawn. The 1:1 line is always clear, the actual line of the study clearly helps to analyze the deviation of the results. This is done in Figure 10 and the deviations and their interpretation are better observed.

3.- In the tables, the use of bold marking in texts in figures 7 to 10 should be clarified.

 4.- Some grammatical corrections, which however the authors should extend in correction in case there are more not detected by this reviewer:

- Line 47 improv and should be improve

- Line 65 great and should be greater

Author Response

(The authors gave the same response as above.)

Round 2

Reviewer 5 Report

This paper can be published after minor revision. 

I indicated in the specific comments the corrections to be made and I invite the authors to examine them very carefully especially those related to the use of acronyms in captions of tables and figures.

Author Response

Thank you for your positive comments and suggestions on our manuscript. We have carefully revised the article through your comments to make more progress. Please refer to the document "Response to Review Comments" for the specific responses.
